# Memory phototransistors based on exponential-association photoelectric conversion law

Zhibin Shao [1], Tianhao Jiang[1], Xiujuan Zhang[1], Xiaohong Zhang [1], Xiaofeng Wu[1], Feifei Xia[2], Shiyun Xiong[1], Shuit-Tong Lee[1] & Jiansheng Jie [1]

Ultraweak light detectors have wide-ranging important applications such as astronomical observation, remote sensing, laser ranging, and night vision. Current commercial ultraweak light detectors are commonly based on a photomultiplier tube or an avalanche photodiode, and they are incompatible with microelectronic devices for digital imaging applications, because of their high operating voltage and bulky size. Herein, we develop a memory phototransistor for ultraweak light detection, by exploiting the charge-storage accumulative effect in CdS nanoribbon. The memory phototransistors break the power law of traditional photodetectors and follow a time-dependent exponential-association photoelectric conversion law. Significantly, the memory phototransistors exhibit ultrahigh responsivity of $3.8 \times 10^9$ A W$^{-1}$ and detectivity of $7.7 \times 10^{22}$ Jones. As a result, the memory phototransistors are able to detect ultraweak light of 6 nW cm$^{-2}$ with an extremely high sensitivity of $4 \times 10^7$. The proposed memory phototransistors offer a design concept for ultraweak light sensing devices.

[1] Institute of Functional Nano & Soft Materials (FUNSOM), Jiangsu Key Laboratory for Carbon-Based Functional Materials & Devices, Soochow University, Suzhou, Jiangsu 215123, P. R. China. [2] School of Chemical and Environmental Engineering, Jiangsu University of Technology, Changzhou, Jiangsu 213001, P. R. China. These authors contributed equally: Zhibin Shao, Tianhao Jiang, Xiujuan Zhang. Correspondence and requests for materials should be addressed to X.Z. (email: xiaohong_zhang@suda.edu.cn) or to S.-T.L. (email: apannale@suda.edu.cn) or to J.J. (email: jsjie@suda.edu.cn)

Detection of ultraweak light enables night vision, remote sensing, astronomical observation, and laser ranging[1–5]. The current commercial ultraweak light detection is normally based on a photomultiplier tube or an avalanche photodiode to achieve the amplification of output electrical signals by multiplying the photogenerated carriers. However, these devices are not compatible with microelectronic devices in digital imaging applications, because of their high operating voltage of 100–1000 V and bulky size[6,7]. Recently, photodetectors based on low-dimensional nanostructures have attracted intense attention due to their unique electronic and optoelectronic properties, as well as high compatibility with microelectronic industry[8–15]. The detectivity, a key figure-of-merit parameter to evaluate the ability of a photodetector to detect ultraweak light, can be enhanced by increasing photoelectric conversion efficiency and/or suppressing dark current in nanostructure-based photodetectors[16,17]. Since a photon normally only excites one pair of electron and hole, thus the generated photocurrent $I_{ph}$ depends on the incident light intensity $P$ and follows the power law $I_{ph} \sim P^{\alpha}$, where the power exponent $\alpha$ is usually less than 1 due to charge recombination at the defective states in actual devices[18,19]. As $P$ increases, $I_{ph}$ shows a sublinear (for actual photodetectors with $\alpha < 1$) or linear increase (for ideal photodetectors with $\alpha = 1$). Nevertheless, in theory even an ideal photodetector ($\alpha = 1$) still suffers from low sensitivity under ultraweak light irradiation.

As a useful complementary to conventional single-function devices, integrated devices that combine two or more different functionalities will achieve higher versatility for applications as well as improved device performance. A variety of such devices have been demonstrated recently, such as self-driven photodetectors[20], haptic memory devices[21], photovoltage field-effect transistors[22], light-emitting phototransistors[23], and resonant thermoelectric nanophotonic devices[24]. Memory devices, by which electrical signals are encoded, stored, and retrieved, are widely integrated in consumer electronic equipment such as digital cameras, smartphones, tablet personal computers, and mobile internet devices. Over the past decade, tremendous efforts have been expended to develop flexible, high-density, multilevel, and non-volatile memory devices based on low-dimensional nanostructures[25,26]. With the goal of constructing all-optical logic processing devices, memory devices incorporated with photodetectors, i.e., optoelectronic memory devices, were designed to realize high-performance, multibit storage of optical signals[27–29]. On the other hand, it is well known that in night photography, more photons can be accumulated on negatives through longer exposure, thus resulting in a clearer night photograph. So, in principle, if photodetectors can be incorporated with memory devices, they should also enable high-performance ultraweak light detection by long-term accumulation of optical signals. However, this kind of integrated devices has yet to be reported.

Here, we demonstrate a memory phototransistor (MPT) based on CdS nanostructures. The MPT exploits the charge storage function induced by the surface states of CdS nanoribbon (NR) to improve the capability of ultraweak light detection. The charge-storage accumulative effect exceeds the power law for traditional photodetectors, leading to a remarkable improvement of power exponent $\alpha$ (the theoretical maximum of $\alpha$ is 1) to 9.3 at a weak light intensity of 50 nW cm$^{-2}$. For such a MPT, $I_{ph}$ depends on both $P$ and the pre-irradiation time $t$, exhibiting an exponential-association photoelectric conversion law of $I_{ph} = a \cdot (1 - e^{-b \cdot P \cdot t})$, where a and b are constants. Significantly, the MPT possesses ultrahigh responsivity of $3.8 \times 10^9$ A W$^{-1}$ and detectivity of $7.7 \times 10^{22}$ Jones (Jones = cm Hz$^{1/2}$ W$^{-1}$), which are 3–5 orders of magnitude higher than those of low-dimensional nanostructure-based photodetectors reported thus far[30–36]. The MPT devices are

capable of detecting an ultraweak light of 6 nW cm$^{-2}$ with an extremely high sensitivity of $4 \times 10^7$. Our MPT devices open up new opportunities for ultraweak light detection applications.

## Results

**Optoelectronic memory function of CdS NR phototransistor.** To incorporate the optical detection and charge storage functions, a phototransistor device based on a surface-state-rich CdS NR was constructed on a SiO$_2$ (300 nm)/n$^+$-Si substrate. The CdS NRs exhibit a single-crystalline wurtzite structure with growth orientation of [001]. The width and thickness of the NRs are 0.4–5 μm and 100–150 nm, respectively, while the length of the NRs is up to several hundreds of micrometers (Supplementary Figure 1). As shown in Fig. 1a, b, indium (In, 200 nm) electrodes, showing ohmic contacts to n-type CdS nanostructures, were defined by photolithography as the source and drain contacts, while the Si substrate served as the global back gate. Fig. 1c shows the electrical transfer characteristic of a CdS NR phototransistor measured in the dark at a fixed drain voltage ($V_{DS}$) of 0.6 V after 10 s of light irradiation (190 nW cm$^{-2}$). A cyclic sweeping voltage of ± 60 V is applied at the gate electrode. The source-drain current ($I_{DS}$) of the CdS NR phototransistor monotonously increases/decreases with increasing/decreasing gate voltage ($V_{GS}$) in the positive/negative sweep, which is consistent with a typical n-channel metal-oxide-semiconductor-field-effect transistor (MOSFET). Interestingly, an obvious threshold voltage ($V_{th}$) hysteresis of 30 V appears between the positive and negative sweeping curves, revealing a memory characteristic of the CdS NR phototransistor. In a control experiment, the transfer characteristic measurement was conducted in the dark without pre-irradiation (Supplementary Figure 2a). However, only a small hysteresis of less than 10 V between the negative and positive sweeping curve can be observed. This weaker charge storage phenomenon is triggered by the electrical input signal and has often been observed in semiconductor nanostructure-based transistors due to adsorbed oxygen-induced surface trapping states[37,38]. Considering the remarkable negative offset of the positive sweeping curve after light irradiation, we infer that the photogenerated holes are captured by a storage medium in the CdS NR[39,40]. Therefore, the entire hysteresis behavior in Fig. 1c arises from a combined effect of the photogenerated holes storage by the optical input signal and the charge storage by the electrical input signal.

To further assess the memory behavior of CdS NR phototransistor, the resistance, electron concentration, and mobility of CdS NR were analyzed. In Fig. 1c, the resistances at $V_{GS} = 0$ in positive and negative sweeps are calculated to be $1.8 \times 10^6$ and $1.1 \times 10^{13}$ Ω, respectively, which correspond to the device's low- and high-impedance states, i.e., bistable states. Correspondingly, the electron concentrations ($n_e$) for low- and high-impedance states are deduced to be $3.5 \times 10^{16}$ and $4.5 \times 10^9$ cm$^{-3}$, respectively. The huge difference originates from the accumulative effect of photogenerated electrons under light irradiation, as we will discuss later. In addition, electron mobility of CdS NR transistor is estimated to be 656 cm$^2$ V$^{-1}$ s$^{-1}$ according to the transfer characteristic curve in linear plot (Supplementary Figure 3). The large mobility can greatly facilitate the transport of photogenerated carriers, thus leading to an ultrahigh sensitivity of the device to incident light. Therefore, even under the irradiation of a weak light of 100 μW cm$^{-2}$, $I_{DS}$ of the device increased drastically and became almost independent of the gate voltage, as shown in Supplementary Figure 2b.

Owing to the pronounced memory characteristics, delayed reading of the light-induced electrical output signals could be

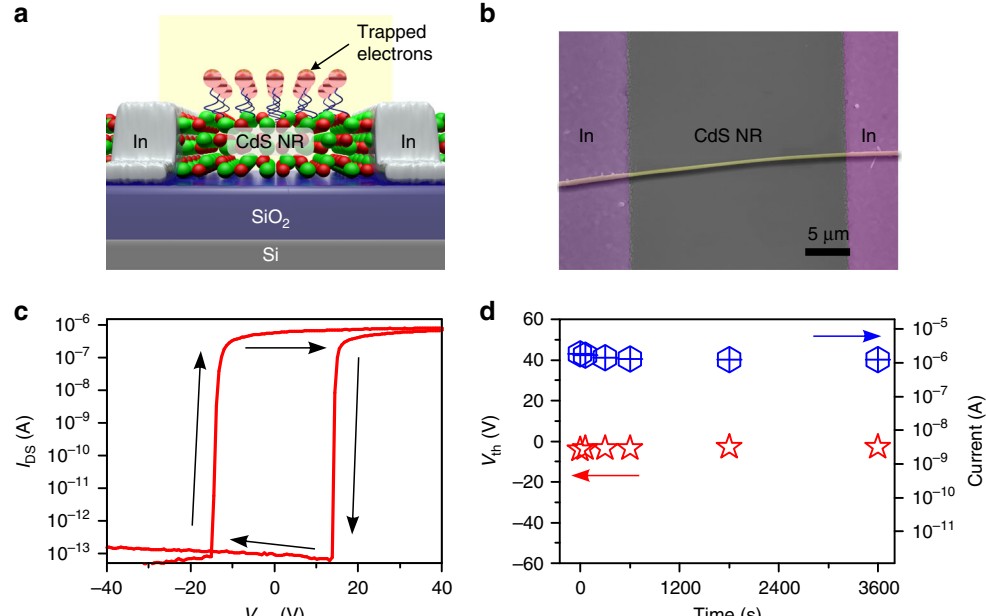

**Fig. 1** Phototransistor device with memory function. **a** Schematic illustration of the CdS NR-based phototransistor device. **b** SEM image of the phototransistor device. **c** The electrical transfer characteristic of the CdS NR phototransistor measured in the dark at a fixed drain voltage of 0.6 V, after it was irradiated with light (190 nW cm$^{-2}$) for 10 s. Black arrows indicate the scanning directions of the curves. **d** Threshold voltage and ON current at different retention times

achieved in the CdS NR phototransistor. Supplementary Figure 4 depicts the electrical transfer characteristics in the positive sweep measured after 0–3600 s delay after light irradiation. From the parameters extracted in Fig. 1d, we note that the threshold voltage and the ON current of CdS NR phototransistor remain almost constant in the delayed reading. Supplementary Figure 5a shows a memory access cycle consisting of optical programing, reading, electrical erasing, and reading. The ON (or OFF) state can be read out at $V_{GS} = 0$ V after light irradiation for 10 s (or electric excitation with a gate bias pulse of 60 V for 2 s). The long retention time of 12,000 s for ON and OFF currents demonstrates the stable storage capability of CdS NR phototransistor (Supplementary Figure 5b, c). Furthermore, Kelvin probe force microscope (KPFM) was utilized to measure the surface potential of CdS NR (Supplementary Figure 6). We note that the surface potential of the NR only decreases slightly after turning off the light, implying that the carrier concentration changes little after light irradiation. This result is consistent with the electrical measurements on the phototransistor.

**Mechanism of optoelectronic memory characteristics**. To gain more insight into the mechanism of optical memory characteristics of the CdS NR phototransistor, a series of electrical measurements were performed for comparison. Fig. 2a depicts the electrical transfer characteristics of the CdS NR phototransistor measured by applying different gate voltages of ± 10 to ± 60 V. It can be seen that as the absolute value of the operating voltage decreases, the reverse sweeping curves are drawn closer to the steady forward sweeping curves, indicating that only electrons are trapped in the storage medium under electric field excitation[41,42]. The Supplementary Figure 7a, b show the electrical transfer characteristics of the CdS NR phototransistor fabricated on the pre-annealed and surface passivated substrates, respectively. Note that the memory characteristics of the devices do not show obvious deterioration. This result excludes the possibilities of defect states on the dielectric layer or adsorbed water

molecules on the substrate as storage mediums. In contrast, Fig. 2b shows that the storage window of the device is significantly reduced after the alumina passivation on the surface of CdS NR via atomic layer deposition (ALD), suggesting that the charge storage sites mainly are the surface states on the CdS NR. Fig. 2c shows the $I_{DS}-V_{DS}$ curves of the low-impedance CdS NR phototransistor plotted in a double logarithmic scale at different temperatures after pre-irradiation. At 300 K, the $I_{DS}-V_{DS}$ curve exhibits a linear ohmic conduction characteristic, indicating the high carrier concentration in the CdS NR and the relatively small potential barrier between the CdS NR and indium electrodes. As the temperature decreases, the resistivity $\rho$ of CdS NR increases (inset of Fig. 2c) and the $I_{DS}-V_{DS}$ curves bend upward at higher voltages. At 22 K, when $V_{DS}$ is in the range of 0–1.2 V, the behavior of $I_{DS}-V_{DS}$ curve is consistent with ohmic conduction (i.e., $I_{DS} \sim V_{DS}$). In contrast, when $V_{DS}$ is in the range of 1.2–10 V, $I_{DS}$ is determined to be proportional to $V_{DS}^{1.4}$, revealing a mixed conduction characteristic between ohmic conduction and space charge limiting conduction ($I_{DS} \sim V_{DS}^2$)[43]. It is known that for an intrinsic CdS NR, the free carriers will decrease dramatically at low temperature, hence the space charges injected from the electrode would dominate in charge transport, giving rise to a space charge limiting current. However, in our work, the amounts of free electrons inside the CdS NR are still very high at 22 K due to the accumulative effect under light irradiation. This is responsible for the mixed transport behavior of carriers in CdS NR.

When oxygen molecules are adsorbed on CdS NR surface, they tend to fill the sulfur vacancies at the surface, leading to the formation of a series of shallow and deep energy states within the band gap of CdS NRs[44–47]. Supplementary Figure 8 shows the electron paramagnetic resonance (EPR) spectrum of CdS NRs. For comparison, the spectra of CdS powder before and after $H_2$/Ar annealing were also measured. The signal at $g = 2.001$ clearly indicates the existence of sulfur vacancies in the CdS NRs due to the high-temperature evaporation process in $H_2$/Ar environment. We note that similar signals from sulfur vacancies were also

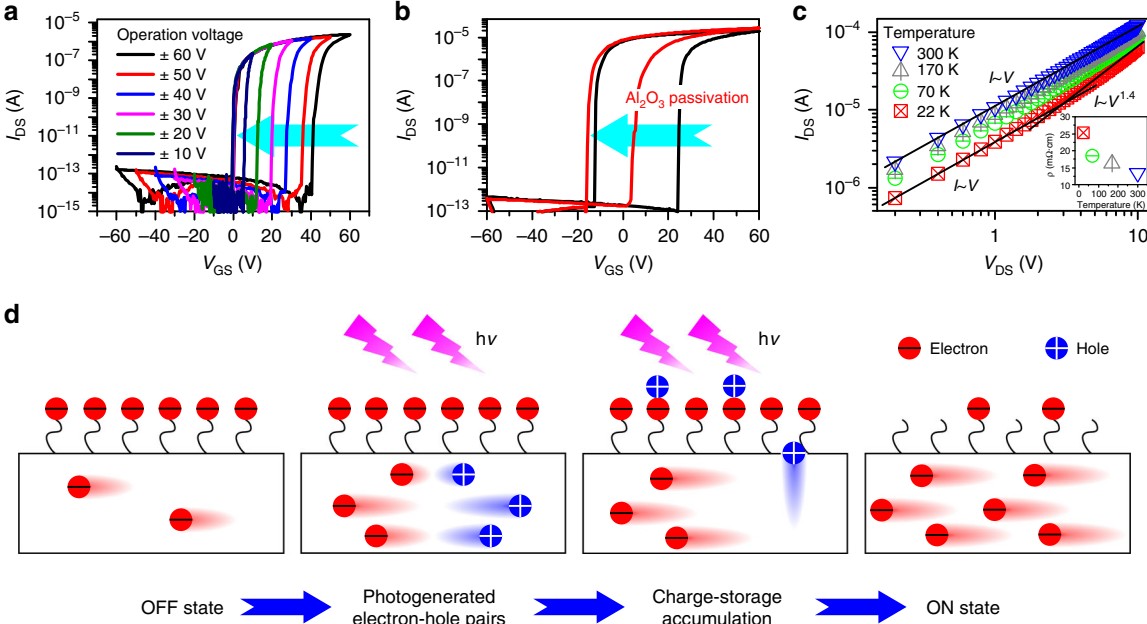

**Fig. 2** Mechanism analysis of optical memory characteristics of CdS NR-based phototransistor. **a** Electrical transfer characteristics of the CdS NR phototransistor measured at different gate voltages from ± 10 to ± 60 V. **b** Electrical transfer characteristics of the CdS NR phototransistor measured before and after $Al_2O_3$ passivation. **c** $I_{DS}-V_{DS}$ curves of the CdS NR phototransistor plotted in a double logarithmic scale at different temperatures. The inset shows the resistivity $\rho$ of the CdS NR at different temperatures. The electrical transfer characteristics in **a** and **b** as well as the $I-V$ curves in **c** were measured in the dark after light pre-irradiation for 10 s (190 nW cm$^{-2}$). **d** Schematic illustration of the optical memory process in the CdS NR phototransistor

observed in the EPR spectra of other metal chalcogen semiconductors[48,49]. Supplementary Figure 9a, b show the optimized configurations of one oxygen molecule-adsorbed CdS surface without and with sulfur vacancy, respectively. It is observed that oxygen molecule has lower adsorption energy ($E_{ads} = -0.87$ eV) on CdS surface with sulfur vacancy, and the molecule is preferentially adsorbed close to the site of sulfur vacancy. The adsorption of oxygen molecule at sulfur vacancy can significantly change the electronic structure of CdS. As shown in Supplementary Figure 9c, a deep energy state is observed above the valence band, which is generated by the adsorbed oxygen molecule according to the projected density of states (PDOS) spectra.

Fig. 2d and Supplementary Figure 10 illustrate the photo-electric storage process in CdS NR phototransistor. In the OFF state, a large amount of electrons will be trapped in the surface states and produce a surface electric field toward NR, which will greatly suppress the amount of free carriers in the device, leading to a low OFF current. Under light irradiation, the photogenerated holes will be rapidly extracted from CdS NR under the surface electric field and subsequently captured by surface states, leaving photogenerated electrons in CdS NR[39,40]. This reduces the recombination probability of photogenerated electron-hole pairs, thus greatly improving the photoelectric conversion efficiency of the device. With increasing irradiation time, the photogenerated holes are continuously injected into the surface storage states of CdS until the trapped electrons are fully recombined with holes. Consequently, the ON current of the device is greatly enhanced by the accumulated electrons within CdS NR. The ultralow dark current, along with the high photoelectric conversion efficiency introduced by surface states, makes the surface-state-rich CdS NR an outstanding candidate for ultraweak light detection. On the other hand, although both optical storage and electrical storage effects contribute to the memory effect of the CdS NR-based phototransistors (Fig. 1c and

Supplementary Figure 2a), the latter capability is much weaker than the former. This is also vital to realize a high-performance photodetection under ultraweak light by avoiding the mutual interference between the electrical and optical input signals (the detailed discussion could be found in Supplementary Figure 10). In contrast, traditional floating-gate optoelectronic memories possess nearly equivalent electrical and optical storage capabilities[39]. Strong interference from electrical signal will limit their performance for ultraweak light detection.

**Performance analysis and optimization of CdS NR MPT**. To assess the detection performance of CdS NR MPTs, transfer characteristics and temporal photoresponse under light irradiation with different intensities were measured. Fig. 3a and Supplementary Figure 11 show the electrical transfer curves of a CdS NR MPT measured in the dark after 10 s of light irradiation with different intensities of 0–2000 nW cm$^{-2}$ in semi-log and linear plots. It can be seen that the curves shift to the upper left as the light intensity increases, and the minimum light intensity that the device can detect is about 50 nW cm$^{-2}$. Further, the photocurrents, threshold voltages, and photodetection performance of the CdS NR MPT can be extracted. Fig. 3b, c show photocurrent versus light intensity ($I_{ph}-P$) characteristics in linear and double logarithmic coordinates, respectively. As is known, for conventional photodetectors, the photocurrent depends on a power function of the incident light intensity, i.e., $I_{ph} \sim P^\alpha$[18,19]. In this case, the $I_{ph}-P$ characteristics can be plotted as a straight line with a slope of $\alpha$ in double logarithmic coordinates. For an ideal photodetector, the power index $\alpha$ is equal to the theoretical maximum of 1. However, $\alpha$ is normally less than 1 due to the recombination of photogenerated carriers introduced by the trap states in the photodetector. However, in Fig. 3c, the slope of the $I_{ph}-P$ characteristic, i.e., $\alpha$, gradually increases with decreasing light intensity. Notably, as shown in

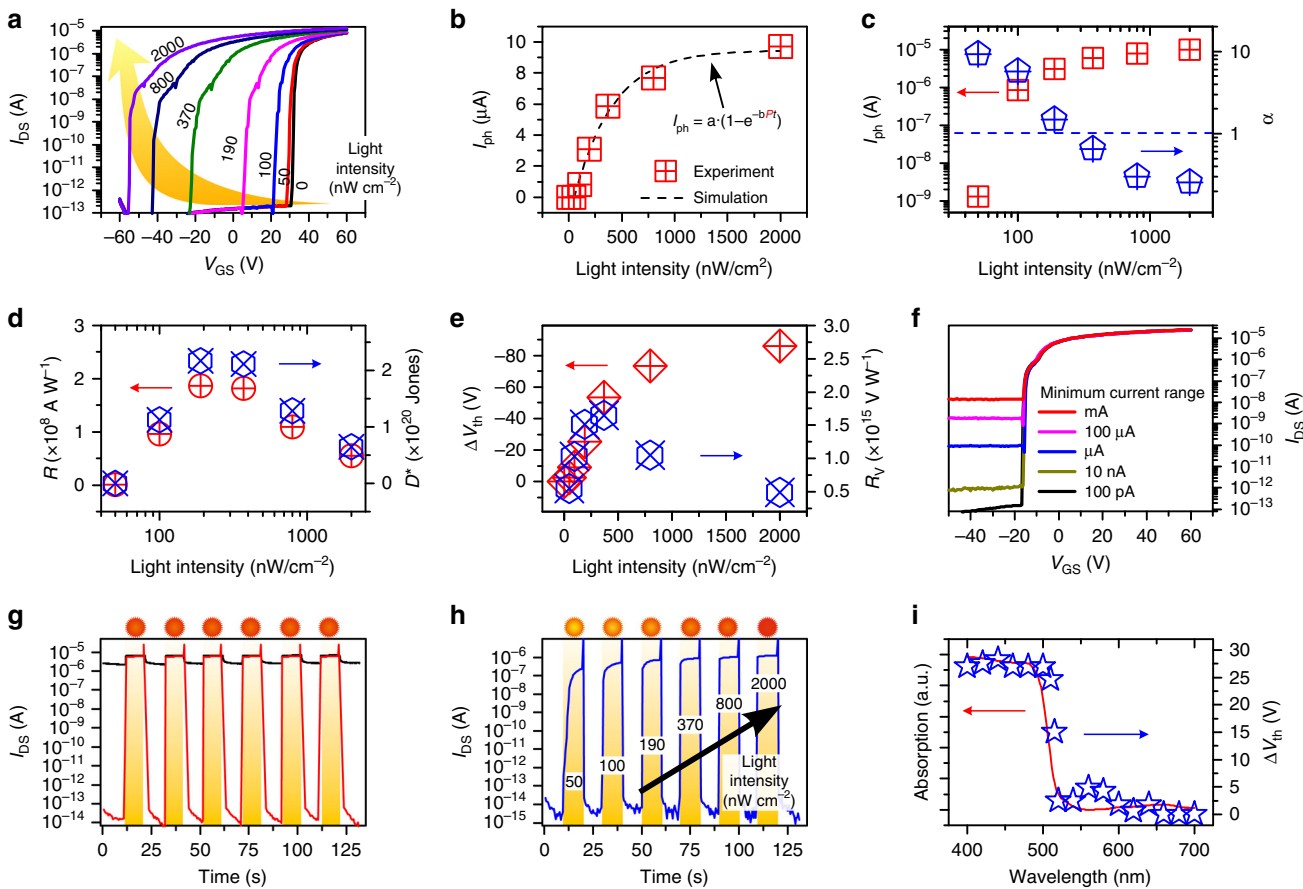

**Fig. 3** Photodetection performance of CdS NR MPT. **a** Electrical transfer characteristics of the CdS NR phototransistor measured in the dark after it was irradiated with different light intensities of 0–2000 nW cm$^{-2}$ for 10 s. $V_{DS}$ was fixed at 0.6 V during measurements. **b** Photocurrent as a function of light intensity in linear coordinate at $V_{GS} = 30$ V. **c** Photocurrent and corresponding α as a function of light intensity in double logarithmic coordinates at $V_{GS} = 30$ V. **d** R, D*, and **e** $\Delta V_{th}$, $R_V$ extracted from (**a**). **f** Transfer characteristics measured in different minimum current ranges of the sourcemeter from 100 pA to 1 mA. **g** Photoresponse curves of the CdS NR phototransistor measured under pulsed light irradiation (2000 nW cm$^{-2}$). A fixed gate voltage (corresponding to black curve) or a periodically pulsed gate voltage (corresponding to red curve) was applied during measurements. **h** Photoresponse curve of the CdS NR phototransistor measured under pulsed light irradiation with a gradually increased light intensity from 50 to 2000 nW cm$^{-2}$. A periodic pulsed gate voltage was applied. **i** Spectral response of the CdS NR phototransistor measured in the wavelength range 400–700 nm, along with the light absorption spectrum of NRs for comparison

Fig. 3c, α exceeds 1 at the light intensity of less than ~200 nW cm$^{-2}$, and even reaches 9.3 at 50 nW cm$^{-2}$, surpassing the theoretical limit of traditional photodetectors. This record high α value is derived from the substantial increase in photoelectric conversion efficiency due to the charge-storage accumulative effect of CdS NR MPTs. However, at a light intensity of higher than ~200 nW cm$^{-2}$, the storage states of the CdS NR are not enough to hold all the photogenerated holes, so that the remaining holes will still recombine with the photogenerated electrons, thereby reducing the photoelectric conversion efficiency and thus the α value.

To clarify the working mechanism of the device, we propose an exponential-association photoelectric conversion law, $I_{ph} = a \cdot (1 - e^{-b \cdot P \cdot t})$, for the MPTs on the basis of the charge accumulation model (see the detailed formula derivation in Supplementary Note 1). In the formula, a represents the saturation value of $I_{ph}$, and b·P represents the rate constant of MPT approaching the saturation value. As shown in Fig. 3b, the photocurrent of the CdS NR MPT at different light intensities can be well fitted by the exponential law. The exponential-association photocurrent curve heads uphill, and finally approaches a plateau. From Fig. 3a, the higher $P$ does accelerate the process of the MPT photocurrent reaching

saturation. Due to the charge-storage cumulative effect and large photoelectric conversion efficiency, the device exhibits an unprecedentedly high performance with a responsivity ($R$) of $10^8$ A W$^{-1}$ and a detectivity ($D^*$) of $10^{20}$ Jones at a light intensity of 190 nW cm$^{-2}$ (Fig. 3d and see the detailed calculation in the Supplementary Note 2), which represent the best values ever reported for the photodetectors based on low-dimensional semiconductors thus far[33,50]. It is worth noting that the maximum values of responsivity and detectivity appear at a light intensity of 190–380 nW cm$^{-2}$ (α ≈ 1). At this point, almost all the electrons at storage states have been recombined with photogenerated holes and the device performance reached the highest. If we assume that the number of surface states is approximately equal to the number of photogenerated electrons in the device channel, the density of surface states could be estimated to be $1.9 \times 10^{12}$–$3.7 \times 10^{12}$ cm$^{-2}$ from the photocurrent at light intensity of 190–380 nW cm$^{-2}$. To gain statistical significance, 30 CdS NR phototransistors were measured after 10 s of light irradiation (190 nW cm$^{-2}$) (Supplementary Figure 12). The consistent device performance indicates the high reproducibility of the CdS NR MPTs. The slight variation of the device performance is mainly due to the different device

surface conditions arising from the fluctuation in NR growth and device construction process.

Current response to the light irradiation is a key quality factor in characterizing the performance of photodetectors. However, to measure the extremely low dark current, a sophisticated current characterization system is normally needed. Alternatively, the shift of the threshold voltage also offers a feasible way to characterize the photoresponse of photodetectors[51,52]. Fig. 3e shows the threshold voltage shift versus the light intensity ($\Delta V_{th}-P$, red symbol) characteristic. The absolute value of the threshold voltage shift gradually increases with the enhancement of light intensity, and tends to saturate at larger light intensity. In Fig. 3e, similar to the current responsivity, the voltage responsivity ($R_V$, blue symbol) of the CdS MPT exhibits a maximum value of $10^{15}$ V W$^{-1}$ at the light intensity of around 200 nW cm$^{-2}$. Fig. 3f shows the transfer characteristics measured in different minimum current ranging from 100 pA to 1 mA of the sourcemeter. The overlapped curves indicate that the voltage responsivity is actually independent of the measurement accuracy of the characterization system. This means a low-accuracy sourcemeter can be used to read the ultraweak light signals in practical application, thus greatly simplifying the detection system.

Real-time high-sensitivity photodetection of MPT can be achieved by applying periodic erase pulses (Supplementary Figure 13). Fig. 3g shows the photoresponse curves of the CdS NR MPT measured at 0 V gate voltage (black curve) and under the application of a periodic pulse gate voltage (red curve). From the magnified photoresponse curve in Supplementary Figure 14, the rise/fall time can be estimated to be 0.12/0.7 and 0.5/0.5 s in the cases of with and without an erase pulse, respectively. Notably, after applying the erase pulse, the OFF current of CdS NR MPT can be greatly suppressed, resulting in an ultrahigh sensitivity of up to ~$10^8$. Due to limited photoelectric conversion efficiency, conventional photodetectors usually exhibit a low sensitivity of less than 10 when detecting an ultraweak light, which will easily lead to erroneous reading of the photoelectric

signal[34,35,53,54]. Fig. 3h shows that the sensitivity of the MPT is still up to ~$10^7$ due to the charge-storage accumulative effect even under an ultralow light intensity of 50 nW cm$^{-2}$. Fig. 3i shows the light absorption curve of CdS NRs, along with the spectral response characteristic of CdS NR MPT for comparison. The overlapped curves demonstrate that the charges stored in the device after light irradiation are derived from the photogenerated carriers in CdS NRs.

To improve the photodetection performance, we fabricated a MPT with a suspended CdS NR across the channel on the substrate. The number of surface storage states could be effectively increased by exposing the bottom surface of the NR[55]. As shown in Fig. 4a, b, a channel with a depth of 200 nm and a width of 8 μm was etched on the SiO$_2$/Si substrate by reactive ion etching (RIE) method. The suspended NR MPT was then fabricated by aligning a CdS NR vertically across the etched channel. The detailed fabrication process is provided in Supplementary Figure 15. Fig. 4c shows the electrical transfer curves of the suspended CdS NR MPT measured in the dark after 10 s of light irradiation under different light intensities of 0–2000 nW cm$^{-2}$. Due to the increased surface storage states, the threshold voltage shift after light irradiation becomes obvious, and the suspended CdS NR MPT is capable of detecting a minimum light intensity of 25 nW cm$^{-2}$. Notably, the voltage responsivities of the suspended MPT are significantly improved compared to the normal MPT (Fig. 4d). The maximum voltage responsivity is as high as $2.5 \times 10^{15}$ V W$^{-1}$ at a light intensity of 200 nW cm$^{-2}$, and the voltage response of a suspended NR MPT under an ultraweak light irradiation (50 nW cm$^{-2}$ for 10 s) is 3.5 times that of a normal device. In addition, compared to the normal device, the dark current of the suspended CdS NR is further reduced from $10^{-13}$ A to $10^{-14}$ A, while the subthreshold swing decreases from 300–600 mV dec$^{-1}$ to 100–300 mV dec$^{-1}$ (Supplementary Figure 16). The reduced dark current, as well as subthreshold swing, are originated from the stronger suppression toward free electrons in the CdS NR by surface electric field. This is beneficial to further reduce the power consumption of MPT in

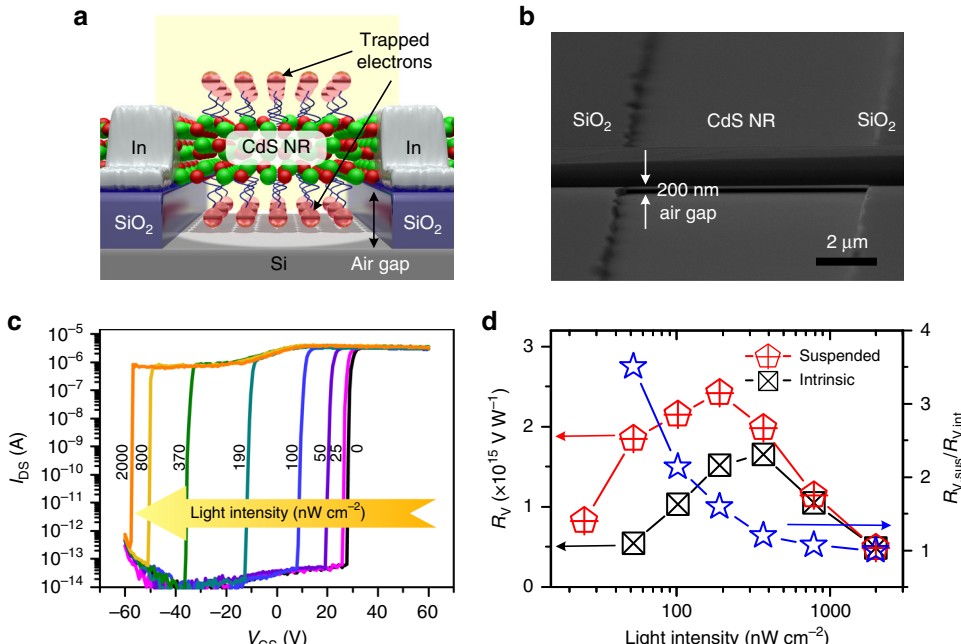

**Fig. 4** Suspended CdS NR MPT. **a** Schematic illustration and **b** SEM image of the suspended CdS NR MPT fabricated above a channel on the substrate. **c** Electrical transfer curves of the suspended CdS NR MPT measured in the dark after 10 s of light irradiation under different light intensities of 0–2000 nW cm$^{-2}$. **d** Voltage responsivities of normal and suspended CdS NR MPT, as well as their ratio at different light intensities

the OFF state[56]. For comparison, if the surface states of CdS NR were reduced by $Al_2O_3$ passivation (Supplementary Figure 17), the threshold voltage shift, the detectable minimum light intensity and voltage responsivity of the passivated CdS NR MPT are clearly deteriorated.

However, further increasing the density of surface states would preferentially increase the electrical memory window, leading to degeneration of the device performance. Electron-beam irradiation is an effective approach to introduce the chalcogen vacancies on the surface of metal chalcogen semiconductor[57–59]. From Supplementary Figure 18, it is observed that the proportion of the electrical storage window in the total storage window increases significantly upon the electron-beam irradiation, manifesting an enhancement of electrical storage capability for the MPT. This result could be attributed to the excess shallow energy states generated after electron-beam irradiation. As discussed in Supplementary Figure 10, the electrical and optical storage capabilities are mainly associated with the shallow and deep energy states, respectively. Due to the small energy barrier between the shallow energy states and conduction band of CdS NR, the electrical storage capability is preferentially improved when the densities of shallow and deep energy states are simultaneously increased. The strong electrical storage will

interfere with the optical storage process, thus limiting the performance of MPT for ultraweak light detection. Therefore, in a MPT, rationally controlling surface state density of the semiconductor nanostructures is crucial for achieving a high performance for weak light detection.

It is noteworthy that the photodetection performance of the CdS MPT strongly depends on the light irradiation time, since more charges could be stored in the device by extending the irradiation time. Fig. 5a depicts the electrical transfer curves of the CdS NR MPT measured in the dark after light irradiation (25 nW cm$^{-2}$) with different time from 0–32,000 s. We note that, as the irradiation time is prolonged, the curve will move to the upper left. Correspondingly, photocurrents and threshold voltage shifts, which are extracted from the electrical transfer curves, increase with increasing irradiation time (Fig. 5b). We note that the time-dependent photocurrent of the CdS MPT can be well fitted with our derived exponential law, $I_{ph} = a \cdot (1 - e^{-b \cdot P \cdot t})$. On the other hand, as the irradiation time increases from 500 to 32,000 s, the photocurrent responsivity and detectivity have increased by a factor of ~500, while the voltage responsivity has increased by more than 15 times (Fig. 5c). Notably, in Fig. 5d, an extremely weak light of 6 nW cm$^{-2}$ can be clearly detected by irradiating the normal and suspended CdS

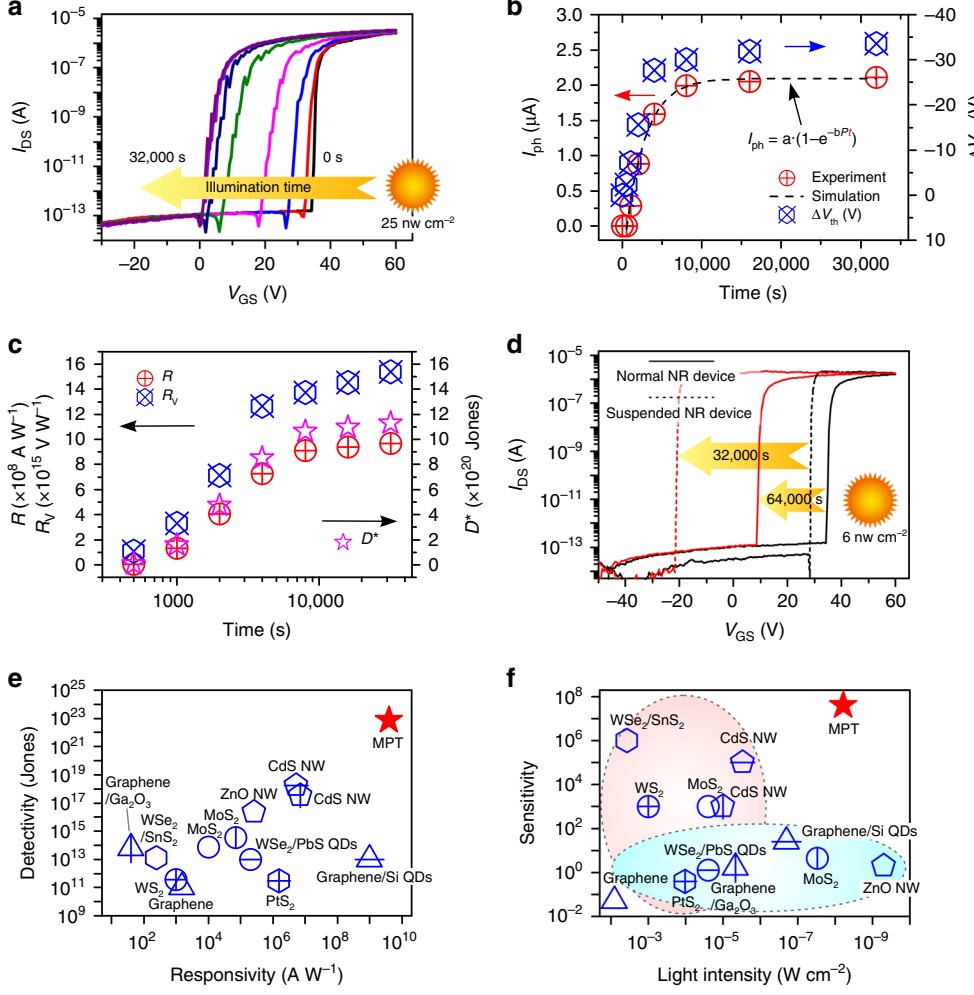

**Fig. 5** Charge-storage accumulative effect. **a** Electrical transfer curves of the CdS NR phototransistor measured in the dark after 0–32,000 s of light irradiation (25 nW cm$^{-2}$). **b** Photocurrents at $V_{GS} = 34$ V, threshold voltage shifts, and **c** photocurrent responsivity, voltage responsivity, and detectivity at different irradiation time, which were extracted from the electrical transfer curves in **a**. **d** Electrical transfer curves of the normal and suspended CdS NR phototransistors after the irradiation by an ultraweak light of 6 nW cm$^{-2}$ for a longer time of 64,000 and 32,000 s, respectively. **e** Responsivity, detectivity, and **f** light intensity, sensitivity of several typical state-of-the-art low-dimensional semiconductor-based photodetectors

MPT for a longer time of 64,000 and 32,000 s, respectively. In this case, a record high photocurrent responsivity of $3.8 \times 10^9$ A W$^{-1}$ and detectivity of $7.7 \times 10^{22}$ Jones are obtained, which are 3–5 orders of magnitude higher than those reported for the traditional photodetectors, as compared in Fig. 5e and Supplementary Table 1[30–36,38,50,53,54,60,61]. The CdS MPT also exhibits a high voltage responsivity of $9.5 \times 10^{16}$ V W$^{-1}$, which is higher than the best values reported for low-dimensional nanostructure-based photodetectors[62]. For the traditional photodetector, a trade-off has to be made between light intensity and sensitivity, as shown in Fig. 5f and Supplementary Table 1[30–36,38,50,53,54,60,61]. In contrast, the MPT enables the detection of the ultraweak light (6 nW cm$^{-2}$) with an ultrahigh sensitivity of $4 \times 10^7$. It is worth noting that the photodetection performance of the MPT is also much better than that of thin film and bulk crystal-based photodetectors (Supplementary Table 2)[63–69].

## Discussion

In summary, we have designed a MPT device that integrates memory and photodetector devices into a single CdS NR phototransistor for ultraweak light detection. Based on the charge-storage accumulative effect, we propose an exponential law ($I_{ph} = a \cdot (1 - e^{-b \cdot P \cdot t})$) in place of the traditional power law ($I_{ph} \sim P^\alpha$) for low-dimensional nanostructure-based photo-detector devices, which achieved a great improvement in photoelectric conversion efficiency. The MPT based on a suspended CdS NR was fabricated to improve device performance via increasing the number of surface storage states. Significantly, the MPT devices exhibit a high responsivity of $3.8 \times 10^9$ A W$^{-1}$ and detectivity of $7.7 \times 10^{22}$ Jones, which are 3–5 orders of magnitude higher than that of previously reported low-dimensional nanostructure-based photodetectors. As a result, the MPT devices can detect an ultraweak light of 6 nW cm$^{-2}$ with an ultrahigh sensitivity of $4 \times 10^7$, which is remarkably higher than the sensitivity (<10) of all previously reported photo-detectors at this light intensity range. In addition to CdS, the deep energy states are often observed in low-dimensional semiconductor nanostructures, such as CdSe, ZnO, GaN, MoS$_2$ etc[70–72]. In this work, we also realized a MPT device based on a CdSe NR, revealing the good generality of our strategy (Supplementary Figure 19). Therefore, MPT devices based on semiconductor nanostructures may be generally achieved through careful tuning of the deep energy surface states, which is beneficial to realize ultraweak light detection at different wavelengths. Our MPT devices overcome the performance bottleneck of low-dimensional nanostructure-based photodetectors, thus presenting a significant advance towards the development of ultraweak light sensors.

## Methods

**Synthesis and characterization of CdS NRs**. Synthesis of CdS NRs was performed by physical vapor deposition in a horizontal quartz tube furnace. Briefly, 0.5 g of CdS powder (Alfa Aesar, 99.999%) was placed on an alumina boat in the center of the tube furnace, then the Si substrates covered with the Au catalyst layer (10 nm) was placed on both sides of the powder at a distance of 7 cm from the powder. After evacuating the tube to a pressure of 0.5 mbar, a high-purity mixture gas of Ar and H$_2$ (5%) was introduced into the tube. When the chamber pressure reached 0.1 bar, the intake and exhaust valves at both ends of the quartz tube were closed. The CdS powder was then rapidly heated to 950 ºC at a rate of 20 ºC min$^{-1}$. After two hours of growth, the yellow product was collected from the Si substrates. Scanning electron microscope (SEM, Hitachi SU5000), X-ray diffraction (XRD, PANalytical Empyrean), atomic force microscope (AFM, Cypher S) and transmission electron microscope (TEM, FEI Tecnai G2 F20 S-TWIN) were used to characterize the morphologies and structures of CdS NRs. EPR measurement was performed (9.07 GHz microwave frequency and 1 mW microwave power) at room temperature by using JEOL JES-FA200.

**Computation of oxygen molecule-adsorbed CdS**. The geometry optimization and the electronic structure calculations were performed based on the first-principles method implemented in the Vienna Ab Initio Simulation Package (VASP). We utilized the generalized gradient approximation (GGA) for the exchange-correlation functional as proposed by Perdew, Burke, and Ernzerhof (PBE). Meanwhile, the DFT-D2 method of Grimme was employed to describe the van der Waals interactions between the O$_2$ and CdS. A plane-wave cutoff energy of 550 eV for the wavefunctions was set, and $8 \times 8 \times 1$ k-points with the Monkhorst–Pack scheme in the first Brillouin zone was employed in the present work. Both the cutoff energy and k grid were tested to be converged in the total energy. The above two layers of models were relaxed in all of the structure optimizations. The convergence criteria for geometric optimization and energy calculation were set to $2.0 \times 10^{-5}$ eV atom$^{-1}$, 0.02 eV Å$^{-1}$, 0.005 Å and $2.0 \times 10^{-6}$ eV atom$^{-1}$ for the tolerance of energy, maximum force, maximum ionic displacement, and self-consistent field (SCF), respectively.

**Device fabrication and performance measurements**. CdS NR-based MPT devices were fabricated as follows: The as-synthesized CdS NRs were dispersed on SiO$_2$ substrate with an approximately ordered orientation via contact printing (Supplementary Figure 20). Then indium (200 nm) electrodes on CdS NRs with 20 μm spacing were defined by photolithography (MJB4, SUSS MicroTec), metal evaporation (Kurt J Lesker Company, NANO 36), and lift-off process. Photolithography was performed using a positive photoresist (Allresist AR-P 5350). To optimize the detection performance of MPTs, a suspended MPT device was fabricated on SiO$_2$ substrate with a surface channel. The surface channel with a depth of 200 nm and a width of 8 μm was first etched on the SiO$_2$/Si substrate by reactive ion etching (RIE). The CdS NR was then aligned vertically across the etched channel. Afterward, indium electrodes were defined on the CdS NR by photolithography, metal evaporation, and lift-off process. The morphologies of the normal and suspended CdS NR-based MPT devices were characterized by SEM. The photoelectric characteristics of MPT devices were detected by a semiconductor characterization system (Keithley 4200-SCS) in the dark or under the light irradiation of a Xe lamp (Beijing Au-light Co. CEL-HXF300). The white light of ~1 mW cm$^{-2}$ was extracted from the Xe lamp by using an optical fiber. The light spot size is around 5 cm, which is far larger than the size of the CdS NR, ensuring the uniform irradiation of light on the device. The light intensity was further weakened by using optical attenuators, and the exact light intensity on the device was calibrated with a light meter (Newport 843-R).

In this work, two key figure-of-merit parameters, i.e., responsivity ($R$) and detectivity ($D^*$), were used to evaluate the performance of CdS NR MPTs[65]. Responsivity represents the photoelectric conversion efficiency of MPTs and is given by the following equation: $R = I_{ph}/P \cdot A$, where A is the area of the device. Detectivity indicates the capability of MPT to detect weak optical signals and can be calculated from the following expression: $D^* = (A^{1/2} \cdot R)/(2qI_d)^{1/2}$, where q is the elementary charge and $I_d$ the dark current. Similar to the current responsivity, the voltage responsivity can be also utilized to evaluate the photoresponse of a phototransistor. It is defined by the following formula: $R_V = \Delta V_{th}/P \cdot A$, where $\Delta V_{th}$ is the shift of threshold voltage.

## Data availability

The data that support the findings of this study are available from the article and Supplementary Information files, or from the corresponding author upon reasonable request.

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

## Acknowledgements

This work was supported by the National Basic Research Program of China (Nos. 2016YFA0202400, 2016YFB0401002), the National Natural Science Foundation of China (Nos. 91833303, 51401138, 51672180, 51622306, 51821002, 21673151, 21703087), China Postdoctoral Science Foundation (No. 2016M601880, 2017T100396), Qing Lan Project, 111 project, Collaborative Innovation Center of Suzhou Nano Science and Technology (NANO-CIC), and the Priority Academic Program Development of Jiangsu Higher Education Institutions (PAPD).

## Author contributions

Z.S., X.J.Z., X.H.Z. and J.J. conceived and designed the experiments. Z.S., X.J.Z., T.J. and X.W. performed the synthesis of CdS NR, device fabrication, characterization, and photoelectric performance measurements. F.X. and S.X. carried out the computation of oxygen molecule adsorbed CdS. Z.S., X.J.Z., S.-T.L and J.J. wrote the paper with input from the other authors. J.J., X.H.Z. and S.-T.L. supervised the project. All the authors contributed to the scientific discussion.

## Additional information

**Competing interests:** The authors declare no competing interests.

