## [Peer Review File · Nature Communications]

Reviewers' comments:

Reviewer #1 (Remarks to the Author):

Title: Memory Photodetectors Based on Exponential-Association Photoelectric Conversion Law

Reviewer comments: This paper describes a novel concept and realization of memory photodetectors (MPDs) using the charge-storage accumulative effect in CdS nanoribbon phototransistors. The CdS phototransistors with the surface states show a high performance with an ultrahigh responsivity and detectivity. These results are great significance in the field of optoelectronic devices.

Some comments are listed below:

1. Why is CdS selected as a material for the MPD? Conventional semiconductors such as Si or Ge can also be considered as a candidate and why?
2. Device performance is mainly compared to low dimensional semiconductor material based devices. Can we also compare with devices based on other materials or dimensions?
3. Please add reference about mobility of CdS nanoribbon (NR) in page 5.
4. It seems that the surface states on the CdS NR are the critical player for this work. Can the authors characterize the surface states? It is missing in the manuscript.
5. CdS NRs were synthesized by PVD in a horizontal quartz tube furnace. It is stated that as-synthesized CdS NRs were uniformly dispersed on SiO₂ substrates via contact printing. Does it mean that as-synthesized CdS NRs were originally dispersed randomly and displaced to uniform position through any printing method? Please provide any SEM images to explain this process.
6. What is the beam spot size of the laser and how do you focus it on the single CdS NR? Please provide detailed information regarding the area of the device, spot size of the laser, and its calculation for responsivity, photoelectric conversion efficiency, and others.

Reviewer #2 (Remarks to the Author):

The manuscript authorized by Shao et al. reports a high-performance photodetector built on CdS nanoribbon. Extremely high responsivity, detectivity and sensitivity have been demonstrated. The high performance originates from the carrier accumulation. A new photoelectric conversion law is developed to describe the experimental results. The referee suggests major revisions based on the following comments before it can be accepted.

1. The demonstrated photodetection is for visible light. Current commercial technique for visible light, especially for the imaging applications, is good enough. Based on the carrier accumulation at the surface states, is it possible to overcome the limitation of the band gap of CdS and extend the response range to IR?
2. The experimental data can be fitted well by the developed law of $I_{ph} = [a \cdot (1 - e)^{-b \cdot P \cdot t}]$. However, the derivation process is questionable, especially "the number of holes captured by surface electrons $\Delta N_h(t)$ should be proportional to the number of photogenerated holes $(A \cdot P \cdot \Delta t)$ in the CdS NR and the number of remaining electrons $(N - N_h(t))$ in the surface states, which can be given by $\Delta N_h(t) = N_h(t + \Delta t) - N_h(t) = B \cdot (A \cdot P \cdot \Delta t) \cdot (N - N_h(t))$, where B is a constant" is not so precise.
3. All the explanation is based on the existence of surface trap states. However, the verification and investigation on the surface states are absent either experimental or theoretically. Where the surface states originate from? Is it dangling bond related? Are the states negatively charged or

positively charged? What's the density of the states?

4. The carrier accumulation mechanism indicates that the response time would be slow. Authors use pulsed light to excite the device and get a relatively faster response. What's the response time? The response speed is important for the photosensors used in the optical communication application. Does this device have this potential?

5. Do any other trap states contribute to the performance? Is surface state the only contributor? If the surface state is the dominant factor, does this mean the thinner or narrower nanoribbons would have better performance?

Reviewer #3 (Remarks to the Author):

1. The work presented by the Author has a wider applicability because of its hybrid nature and has a two-fold approach and is therefore quite interesting for the researcher.

2. The work seems to be original, however, characterization of CdS nano ribbons has not been given in the submitted manuscript.

3. Photoelectric storage Mechanism has been explained, however, a control experiment may be carried out as a function of sulfur vacancies during the synthesis.

4. It would be appreciable and helpful for the author to carry out the conductive AFM studies in the Low and high resistance state of the CDS NRs after and before the light illumination for further understanding the mechanism.

5. Abstract is clear and understandable.

6. May be accepted after minor suggested revision.

Regards

Dr. Bharti Singh

Response to reviewer #1:

Reviewer's comments: This paper describes a novel concept and realization of memory photodetectors (MPDs) using the charge-storage accumulative effect in CdS nanoribbon phototransistors. The CdS phototransistors with the surface states show a high performance with an ultrahigh responsivity and detectivity. These results are great significance in the field of optoelectronic devices.

Reply: We would like to thank reviewer for the very positive comment on our manuscript. Our responses to the reviewer's specific comments are given below:

(1) Why is CdS selected as a material for the MPD? Conventional semiconductors such as Si or Ge can also be considered as a candidate and why?

Reply: CdS was selected as a material for MPDs because of its excellent optoelectronic properties and the existence of deep energy states on the surface. Firstly, CdS nanostructures have been intensively investigated for photodetection due to their superior optoelectronic properties. Secondly, anion vacancies are the primary defects of the high temperature-evaporated II-VI semiconductor nanostructures. The vacancy defects distributed on the surface serve as the adsorption sites of oxidizing gases, especially oxygen in the atmosphere, leading to generation of deep energy states on the surface (Prades et al., **Sens. Actuators B Chem.** 2009, 142, 179). The adsorbed oxygen-induced surface trapping states can serve as charge accumulation centers in MPDs, making CdS nanostructure a promising material for MPDs.

In the MPD devices, to realize high-performance ultraweak-light photodetection, it is vital to avoid the mutual interference between the electrical and optical input signals. To achieve this, semiconductor materials should possess a weak electrical storage capability but a strong optical storage capability. Due to the large energy barrier between surface states and conduction band, the electrical programming process of extracting electrons from the deep energy states is greatly suppressed. As a result, the deep energy states on the surface present weaker electrical storage capability compared with the conventional floating-gate storage mediums.

Persistent photoconductivity (PPC) phenomenon is a persistent process of slowly releasing the photogenerated charges from traps, that is, the inverse of the charge accumulation process. PPC phenomena have been widely observed in semiconductor materials that have deep energy states on the surface, such as II-VI compounds (ZnO, Lee et al., **Adv. Mater.** 2017, 29, 1700951; CdS, Yin et al., **Phys. Rev. Mater.** 2018, 2, 084602), III-V compounds (GaN, Leonard, et al., **Nano Lett.** 2015, 15, 8129), transition metal dichalcogenides (MoS₂, Di Bartolomeo et al., **Adv. Funct. Mater.** 2018, 28, 1800657), and perovskite oxides (SrTiO₃, Liu et al., **Adv. Mater.** 2016, 28, 7768). Therefore, these materials can be considered as potential candidates for MPD

application. As shown in the Supplementary Fig. 19, we have also fabricated a MPD device based on a CdSe NR.

As for Si and Ge, based on previous reports, their PPC behaviors are relatively weak, implying few effective deep energy states exist on their surfaces. Therefore, to fabricate MPDs based on Si and Ge nanostructures, appropriate surface modifications are needed to increase the amount of effective deep energy states. It has been reported that surface/interface treatments, such as surface roughening and interface decoration, are able to improve the PPC behaviors in Si and Ge (Feng et al., *Nano Lett.* 2009, 9, 3453; Rasool et al., *Appl. Phys. Lett.* 2012, 101, 253104).

Accordingly, we added the following discussion in page 15: “In addition to CdS, the deep energy states are often observed in low-dimensional semiconductor nanostructures, such as CdSe, ZnO, GaN, MoS₂ etc.⁷⁰⁻⁷² In this work, we also realized a MPD device based on a CdSe NR, revealing the good generality of our strategy (Supplementary Fig. 19). Therefore, MPD devices based on semiconductor nanostructures may be generally achieved through careful tuning of the deep energy surface states, which is beneficial to realize ultraweak light detection at different wavelengths.”

Supplementary Fig. 19 (a) SEM image of the CdSe NR-based MPD device. (b) Electrical transfer characteristics of the CdSe NR MPD measured in the dark at a fixed drain voltage of 0.6 V, with (red curve) and without (dark curve) 10 s light pre-irradiation ($2 \mu\text{W cm}^{-2}$).

(2) Device performance is mainly compared to low dimensional semiconductor material based devices. Can we also compare with devices based on other materials or dimensions?

Reply: We thank the reviewer for the valuable suggestion. Following reviewer’s suggestion, we have compared the device performance of our MPD device with thin film and bulk crystal-based photodetectors (Supplementary Table 2). The

photodetection performance of MPD is also much better than these devices. Accordingly, we added the Supplementary Table 2 in the Supplementary Information, and the following discussion was added in Page 15: “It is worth noting that the photodetection performance of the MPD is also much better than that of thin film and bulk crystal-based photodetectors (Supplementary Table 2).⁶³⁻⁶⁹”

Supplementary Table 2. Comparison of device performance of the CdS NR MPD with other thin film and bulk crystal-based photodetectors.

Photodetectors	Responsivity (A W ⁻¹)	Detectivity (Jones)	Sensitivity @ light intensity (W cm ⁻²)	Reference
CdS film	7.3×10 ⁵	3.5×10 ¹⁶	1.0 @ 1.9×10 ⁻⁹	1
ZnO film	1.0×10 ⁴	5.2×10 ¹²	0.67 @ 3×10 ⁻³	2
CH ₃ NH ₃ PbI ₃ film	81	10 ¹¹	17 @ 0.3×10 ⁻⁶	3
CsPbBr ₃ film	0.18	6.1×10 ¹⁰	8×10 ³ @ 10 ⁻³	4
ZnSe bulk crystal	4.4	1.4×10 ¹¹	4×10 ³ @ 3	5
CH ₃ NH ₃ PbI ₃ bulk crystal	953	3.2×10 ¹⁴	223 @ 1.8×10 ⁻³	6
MoS ₂ film/Si bulk crystal	300	10 ¹³	8×10 ³ @ 10 ⁻³	7
CdS NR MPD	3.8×10⁹	7.7×10²²	10⁷ @ 6×10⁻⁹	Our work

References

1. Lin, K.-T., et al. Nanocrystallized CdS beneath the surface of a photoconductor for detection of UV light with picowatt sensitivity. *ACS Appl. Mater. Interfaces* **6**, 19866-19875 (2014).
2. Roul, B., et al. Enhanced UV photodetector response of ZnO/Si with AlN buffer layer. *IEEE Trans. Electron Devices* **64**, 4161-4166 (2017).
3. Hu, W., et al. High-performance flexible photodetectors based on high-quality perovskite thin films by a vapor-solution method. *Adv. Mater.* **29**, 1703256 (2017).
4. Li, X. M., et al. Healing all-inorganic perovskite films via recyclable dissolution-recrystallization for compact and smooth carrier channels of optoelectronic devices with high stability. *Adv. Funct. Mater.* **26**, 5903-5912 (2016).
5. Sirkeli, V. P., et al. Enhanced responsivity of ZnSe-based metal-semiconductor-metal near-ultraviolet photodetector via impact ionization. *Phys. Status Solidi RRL* **12**, 1700418 (2017).
6. Lian, Z. P., et al. High-performance planar-type photodetector on (100) facet of MAPbI₃ single crystal. *Sci. Rep.* **5**, 16563 (2015).

7. Wang, L., et al. MoS₂/Si heterojunction with vertically standing layered structure for ultrafast, high-detectivity, self-driven visible-near infrared photodetectors. *Adv. Funct. Mater.* **25**, 2910-2919 (2015).

(3) Please add reference about mobility of CdS nanoribbon (NR) in page 5.

Reply: In this work, electron mobility of CdS NR is estimated according to the transfer characteristic curve plotted in linear coordinate (Supplementary Fig. 3). The mobility (μ) can be estimated by the formula of $\mu = g_m L / W C_0 V_{DS}$, where $g_m = \Delta I_{DS} / \Delta V_{GS}$ is transconductance at a given V_{GS} , L is the channel length, W is the channel width, and C_0 is the gate capacitance per unit area. From the μ vs. V_{GS} curve in Supplementary Fig. 3, a maximum electron mobility value of $656 \text{ cm}^2 \text{ V}^{-1} \text{ s}^{-1}$ is deduced.

Accordingly, we added the Supplementary Fig. 3 in the Supplementary Information, and the discussion in Page 5 was amended as the following: “In addition, electron mobility of CdS NR transistor is estimated to be $656 \text{ cm}^2 \text{ V}^{-1} \text{ s}^{-1}$ according to the transfer characteristic curve in linear plot (Supplementary Fig. 3).”

Supplementary Fig. 3 Electrical transfer characteristic in a linear plot (black line) and the extracted field-effect mobility as a function of V_{GS} (blue line) of the CdS NR transistor. The device is measured in the dark at a fixed drain voltage of 0.6 V.

(4) It seems that the surface states on the CdS NR are the critical player for this work. Can the authors characterize the surface states? It is missing in the manuscript.

Reply: We thank the reviewer for the valuable suggestion. We note that direct characterization of the surface states is difficult through conventional characterization techniques. Therefore, as an alternative, we first confirmed the existence of sulfur

vacancies in the CdS NRs by electron paramagnetic resonance (EPR) spectroscopy, and then the surface states induced by oxygen adsorption at sulfur vacancies were studied by the first-principles calculation. Supplementary Fig. 8 shows the EPR spectrum of the as-prepared CdS NRs. For comparison, the spectra of CdS powder before and after H₂/Ar annealing were also measured. The signal at g=2.001 clearly indicates the existence of sulfur vacancies in the CdS NRs due to the high-temperature evaporation process in H₂/Ar environment. It is noted that similar signals from sulfur vacancies were observed in the EPR spectra of other metal chalcogen semiconductors (MoS₂, Zhang et al., *Adv. Funct. Mater.* 2018, 28, 1707578; ZnIn₂S₄, Wang et al., *J. Mater. Chem. A* 2017, 5, 8451).

Supplementary Fig. 8 EPR spectrum of CdS NRs. For comparison, the spectra of CdS powder before and after H₂/Ar annealing at 800 °C for 1 h were also measured.

On the other hand, the first-principles calculation was conducted to study the formation of surface energy states within the band gap of CdS NRs after adsorption of oxygen molecules at sulfur vacancies. Supplementary Fig. 9a and 9b show the optimized configurations of one oxygen molecule-adsorbed CdS surface without and with sulfur vacancy, respectively. It is noteworthy that oxygen molecule has lower adsorption energy ($E_{\text{ads}} = -0.87$ eV) on CdS surface with sulfur vacancy, and the molecule is preferentially adsorbed close to the site of sulfur vacancy. The adsorption of oxygen molecule at sulfur vacancy can significantly change the electrical structure of CdS. As shown in Supplementary Fig. 9c, a new deep energy state is observed above the valence band, which is generated by the adsorbed oxygen molecule according to the projected density of states (PDOS) spectra.

Supplementary Fig. 9 (a,b) Energetically favorable configurations of one oxygen molecule-adsorbed CdS surface without and with sulfur vacancy, respectively. The adsorption energy E_{ads} is defined by the following equation: $E_{ads} = E_{O_2-cds} - E_{cds} - E_{O_2}$, where E_{O_2-cds} , E_{cds} , and E_{O_2} represent the energy of the full adsorption system, the CdS crystal, and the adsorbed O_2 molecule, respectively. (c) Total density of states (DOS, blue lines) spectra of CdS before and after the oxygen adsorption. In the top, the gray and red lines represent the projected density of states (PDOS) of CdS and O_2 in the adsorption system, respectively. The arrow indicates the surface energy state introduced from the adsorbed oxygen molecule.

Accordingly, we added the Supplementary Fig. 8 and Fig. 9 in the Supplementary Information, and the following discussion was changed in Page 7–8: “When oxygen molecules are adsorbed on CdS NR surface, they tend to fill the sulfur vacancies at the surface, leading to the formation of a series of shallow and deep energy states within the band gap of CdS NRs.⁴⁴⁻⁴⁷ Supplementary Fig. 8 shows the electron paramagnetic resonance (EPR) spectrum of CdS NRs. For comparison, the spectra of CdS powder before and after H_2/Ar annealing were also measured. The signal at $g=2.001$ clearly indicates the existence of sulfur vacancies in the CdS NRs due to the high-temperature evaporation process in H_2/Ar environment. We note that similar signals from sulfur vacancies were also observed in the EPR spectra of other metal chalcogen semiconductors.^{48, 49} Supplementary Fig. 9a and 9b show the optimized configurations of one oxygen molecule-adsorbed CdS surface without and with sulfur vacancy, respectively. It is observed that oxygen molecule has lower adsorption energy ($E_{ads} = -0.87$ eV) on CdS surface with sulfur vacancy, and the molecule is preferentially adsorbed close to the site of sulfur vacancy. The adsorption of oxygen molecule at sulfur vacancy can significantly change the electrical structure of CdS. As shown in Supplementary Fig. 9c, a new deep energy state is observed above the valence band, which is generated by the adsorbed oxygen molecule according to the projected density of states (PDOS) spectra.”

The experimental and computational details were added in Page 16–17 in revised manuscript: “EPR measurement was performed (9.07 GHz microwave frequency and 1 mW microwave power) at room temperature by using JEOL JES-FA200.

Computation of oxygen molecule-adsorbed CdS: The geometry optimization and the electronic structure calculations were performed based on the first-principles method implemented in the Vienna Ab Initio Simulation Package (VASP). We utilized the generalized gradient approximation (GGA) for the exchange-correlation functional as proposed by Perdew, Burke, and Ernzerhof (PBE). Meanwhile, the DFT-D2 method of Grimme was employed to describe the van der Waals interactions between the O₂ and CdS. A plane-wave cutoff energy of 550 eV for the wavefunctions was set, and 8×8×1 k-points with the Monkhorst–Pack scheme in the first Brillouin zone was employed in the present work. Both the cutoff energy and k grid were tested to be converged in the total energy. The above two layers of models were relaxed in all of the structure optimizations. The convergence criteria for geometric optimization and energy calculation were set to 2.0×10^{-5} eV atom⁻¹, 0.02 eV Å⁻¹, 0.005 Å and 2.0×10^{-6} eV atom⁻¹ for the tolerance of energy, maximum force, maximum ionic displacement, and self-consistent field (SCF), respectively.”

(5) CdS NRs were synthesized by PVD in a horizontal quartz tube furnace. It is stated that as-synthesized CdS NRs were uniformly dispersed on SiO₂ substrates via contact printing. Does it mean that as-synthesized CdS NRs were originally dispersed randomly and displaced to uniform position through any printing method? Please provide any SEM images to explain this process.

Reply: We thank the reviewer for the careful review. The CdS NRs with random orientations were first grown on the Si substrate (Supplementary Fig. 1a), and then the as-synthesized CdS NRs can be dispersed with an approximately ordered orientation on SiO₂ substrates *via* contact printing (Supplementary Fig. 20b). To further understand the process, a schematic illustration of the contact printing process was provided in Supplementary Fig. 20a. The contact printing method involves the directional sliding of a Si growth substrate on top of a SiO₂/Si substrate. After the sliding step, the CdS NRs are detached from the Si substrate, resulting in the direct transfer of aligned CdS NRs onto the SiO₂/Si substrate (Supplementary Fig. 20b). To facilitate the fabrication of single NR-based device, the density of aligned CdS NRs is controlled by adjusting the interaction force between the two substrates.

Accordingly, we added the Supplementary Fig. 1 & Fig. 20 in the Supplementary Information, and the following experimental details were added in Page 17: “The as-synthesized CdS NRs were dispersed on SiO₂ substrate with an approximately ordered orientation *via* contact printing (Supplementary Fig. 20).”

The following discussion about Supplementary Fig. 20 was added in the Supplementary Information: “The contact printing method involves the directional sliding of a Si growth substrate on top of a SiO₂/Si substrate. After the sliding step, the CdS NRs are detached from the Si substrate, resulting in the direct transfer of aligned CdS NRs to the SiO₂/Si substrate. To fabricate the single NR-based device, the density of aligned CdS NRs is controlled by adjusting the interaction force between the two substrates.”

Supplementary Fig. 1 Characterizations of CdS NRs synthesized by physical vapor deposition. (a) SEM image, (b) EDS spectrum, (c) XRD pattern of the CdS NRs. (d) AFM three-dimensional (3D) morphology image, (e) TEM image, and (f) SAED pattern of an individual CdS NR.

Supplementary Fig. 20 (a) Schematic illustration of contact printing process. (b) SEM image of semi-aligned CdS NRs on SiO₂/Si substrate.

(6) What is the beam spot size of the laser and how do you focus it on the single CdS NR? Please provide detailed information regarding the area of the device, spot size of the laser, and its calculation for responsivity, photoelectric conversion efficiency, and others.

Reply: We didn't use laser as the light source since it is more difficult to focus the laser beam onto the device. Alternatively, a Xe lamp (Beijing Au-light Co. CEL-HXF300) was used as the light source to achieve uniform light irradiation on the device. White light of $\sim 1 \text{ mW cm}^{-2}$ was extracted from the Xe lamp by using an optical fiber. The light spot size is around 5 cm, which is far larger than the size of the CdS NR, ensuring the uniform irradiation of light on the device. The light intensity was further weakened by using optical attenuators, and the exact light intensity on the device was calibrated with a light meter (Newport 843-R).

The area of the device is equal to the area of CdS NR in the device channel. It can be given by $A = L \cdot W$, where L and W are the channel length and width, respectively. In Fig. 3, a device with L and W of 23 and $0.38 \mu\text{m}$, respectively, was measured. In the photodetector, current responsivity that represents the photoelectric conversion efficiency can be evaluated by the following equation: $R = I_{ph}/P \cdot A$. Detectivity indicates the capability of a device to detect weak optical signals and can be estimated from the following expression: $D^* = (A^{1/2} \cdot R)/(2qI_d)^{1/2}$, where q is the elementary charge and I_d is the dark current. Similar to the current responsivity, the voltage responsivity can be also utilized to evaluate the photoresponse of a phototransistor. It is defined by the following formula: $R_v = \Delta V_{th}/P \cdot A$, where ΔV_{th} is the shift of threshold voltage after the light irradiation. Here, we take the calculation at a light intensity of 190 nW cm^{-2} as an example. From the curves in Fig. 3, the dark current and photocurrent at $V_{GS}=30 \text{ V}$ is 0.2 pA and $3.10 \mu\text{A}$, respectively, while the ΔV_{th} is 25.2 V . Based on the above equations and data, responsivity, detectivity, and voltage responsivity of CdS MPD are deduced to be $1.86 \times 10^8 \text{ A W}^{-1}$, $2.18 \times 10^{22} \text{ Jones}$, and $1.52 \times 10^{15} \text{ V W}^{-1}$, respectively.

Accordingly, we amended the following discussion in Page 10: “Due to the charge-storage cumulative effect and large photoelectric conversion efficiency, the device exhibits an unprecedentedly high performance with a responsivity (R) of 10^8 A W^{-1} and a detectivity (D^*) of 10^{20} Jones at a light intensity of 190 nW cm^{-2} (Fig. 3d and see the detailed calculation in the Supplementary Information)”.

We also added the following details in the experimental section in Page 17 & 18: “The photoelectric characteristics of MPD devices were detected by a semiconductor characterization system (Keithley 4200-SCS) in the dark or under the light irradiation of a Xe lamp (Beijing Au-light Co. CEL-HXF300). The white light of $\sim 1 \text{ mW cm}^{-2}$

was extracted from the Xe lamp by using an optical fiber. The light spot size is around 5 cm, which is far larger than the size of the CdS NR, ensuring the uniform irradiation of light on the device. The light intensity was further weakened by using optical attenuators, and the exact light intensity on the device was calibrated with a light meter (Newport 843-R).” and “Similar to the current responsivity, the voltage responsivity can be also utilized to evaluate the photoresponse of a phototransistor. It is defined by the following formula: $R_v = \Delta V_{th}/P \cdot A$, where ΔV_{th} is the shift of threshold voltage.”

The detailed calculation of the device performance for MPD was provided in the Supplementary Information: “In the photodetector, current responsivity R can be evaluated by the following equation: $R = I_{ph}/P \cdot A$, where I_{ph} , P , A are the photocurrent, light intensity, and the area of the device, respectively. The area of the device is equal to the area of CdS NR in the device channel. It can be given by $A = L \cdot W$, where L and W are the channel length and width, respectively. Detectivity D^* indicates the capability of a MPD to detect weak optical signals and can be calculated from the following expression: $D^* = (A^{1/2} \cdot R)/(2qI_d)^{1/2}$, where q is the elementary charge and I_d is the dark current. It is assumed that the dark current is dominated by the shot noise for estimating detectivity. Voltage responsivity R_v can be also utilized to evaluate the photoresponse of a phototransistor. It is defined by the following formula: $R_v = \Delta V_{th}/P \cdot A$, where ΔV_{th} is the shift of threshold voltage after the light irradiation. In Fig. 3, a device with L and W of 23 and 0.38 μm , respectively, was measured. Here, we take the calculation at the light intensity of 190 nW cm^{-2} as an example. From the curves in Fig. 3, the dark current and photocurrent at $V_{GS}=30$ V is 0.2 pA and 3.10 μA , respectively, and ΔV_{th} is 25.2 V. Based on the above equations and data, responsivity, detectivity, and voltage responsivity of the CdS MPD are deduced to be $1.86 \times 10^8 \text{ A W}^{-1}$, 2.18×10^{22} Jones, and $1.52 \times 10^{15} \text{ V W}^{-1}$, respectively.”

Response to reviewer #2:

Reviewer's comments: The manuscript authorized by Shao et al. reports a high-performance photodetector built on CdS nanoribbon. Extremely high responsivity, detectivity and sensitivity have been demonstrated. The high performance originates from the carrier accumulation. A new photoelectric conversion law is developed to describe the experimental results. The referee suggests major revisions based on the following comments before it can be accepted.

Reply: We would like to thank reviewer for the careful review and constructive comments on our manuscript. Our response to reviewer's specific comments is given below:

(1) The demonstrated photodetection is for visible light. Current commercial technique for visible light, especially for the imaging applications, is good enough. Based on the carrier accumulation at the surface states, is it possible to overcome the limitation of the band gap of CdS and extend the response range to IR?

Reply: We agree with the reviewer that the IR detection is very important. Nevertheless, in this work, we didn't observe an obvious photoresponse of the CdS NR MPD to IR light. The possible reasons might be: (i) The weak IR absorption of the surface states. Due to the high penetration depth of IR light in CdS NR, the IR absorption of surface states should be very weak. (ii) The strong carrier recombination rate at the surface. Even if the electron-hole pairs could be generated by the IR light absorption of surface states, the strong surface recombination will cause the fast recombination of the photo-generated carriers at surface, thus contributing little to the photocurrent.

To overcome the band gap limitation of the CdS NR, one possible way is to introduce inter-band defect levels in NR through doping or adding additional IR responsive sensitizers on the NR surface. Another way is the use of semiconductors with different band gaps as channel materials; indeed the deep surface states have been observed in a host of semiconductor nanostructures, such as II-VI compounds (ZnO, Lee et al., **Adv. Mater.** 2017, 29, 1700951; CdS, Yin et al., **Phys. Rev. Mater.** 2018, 2, 084602), III-V compounds (GaN, Leonard, et al., **Nano Lett.** 2015, 15, 8129), transition metal dichalcogenides (MoS₂, Di Bartolomeo et al., **Adv. Funct. Mater.** 2018, 28, 1800657), and perovskite oxides (SrTiO₃, Liu et al., **Adv. Mater.** 2016, 28, 7768). In principle, these nanostructures would have the potential to be used for the fabrication of MPD devices with photoresponse at different wavelengths.

Accordingly, we added the following discussion in page 15: "In addition to CdS, the deep energy states are often observed in low-dimensional semiconductor nanostructures, such as CdSe, ZnO, GaN, MoS₂ etc.⁷⁰⁻⁷² In this work, we also realized a MPD device based on a CdSe NR, revealing the good generality of our strategy

(Supplementary Fig. 19). Therefore, MPD devices based on semiconductor nanostructures may be generally achieved through careful tuning of the deep energy surface states, which is beneficial to realize ultraweak light detection at different wavelengths.”

(2) The experimental data can be fitted well by the developed law of $I_{\text{ph}} = a \cdot (1 - a \cdot (1 - e^{-b \cdot P \cdot t}))$. However, the derivation process is questionable, especially “the number of holes captured by surface electrons $\Delta N_h(t)$ should be proportional to the number of photogenerated holes ($A \cdot P \cdot \Delta t$) in the CdS NR and the number of remaining electrons ($N - N_h(t)$) in the surface states, which can be given by $\Delta N_h(t) = N_h(t + \Delta t) - N_h(t) = B \cdot (A \cdot P \cdot \Delta t) \cdot (N - N_h(t))$, where B is a constant” is not so precise.

Reply: We thank the reviewer for the insightful comment. Indeed, this equation only provides an approximate description on the physical process of carrier accumulation on the surface states. The derivation process is mainly based on the previous studies on the decay behavior of persistent photoconductivity in bulk materials by assuming that the charge accumulation process is the inverse of the decay process. The decay process for persistent photoconductivity can be precisely described by stretched-exponential function (Jiang et al., **Phys. Rev. B** 1989, 40, 10025; Lin et al., **Phys. Rev. B** 1990, 42, 5855):

$$f(t) = f(0)e^{-\left(\frac{t}{\tau}\right)^\beta} \quad (\text{Kohlrusch, } \mathbf{Annalen der Physik}, 1854, 167, 179)$$

where τ is the characteristic decay time constant and β is the decay exponent. During the decay process of persistent photoconductivity, the number of traps reoccupied by electrons can be given by $\Delta N_e(t) = B \cdot N_{fe} \cdot N_{tu}(t)$, where N_{fe} is the number of free electrons in the semiconductor and $N_{tu}(t)$ is the number of traps unoccupied by electrons. Since free electrons in the bulk have to diffuse a long distance to reach the traps at surface/interface, B is a time-dependent variate, causing that the exponential decay is stretched in time by the factor β ($0 < \beta < 1$). The stretched exponential function is extensively used as a phenomenological description of physics relaxation process (Williams et al., **Transactions of the Faraday Society** 1970, 66, 80; Palmer et al., **Phys. Rev. Lett.** 1984, 53, 958). In our work, due to the small thickness of CdS NR and the existence of sufficient surface states, the photogenerated holes do not need to diffuse a long distance to the traps, leading to a nearly time-independent constant B. In this case, the charge accumulation process is an exponential Debye evolution process with $\beta = 1$, a particular case of stretched exponential evolution process (Phillips, **Rep. Prog. Phys.** 1996, 59, 1133). As shown in Fig. 3b and Fig. 5b in the manuscript, the derived exponential function $I_{\text{ph}} = a \cdot (1 - e^{-b \cdot P \cdot t})$ does fit the experimental results well. It is also noted that, for the

case of the existence of few surface states or a large NR thickness, the photogenerated holes may have to diffuse a long distance to the traps. Therefore, in order to accurately describe the charge accumulation process, an exponent factor β should be introduced to correct the derived exponential function.

Accordingly, we added the following discussions in page 3 in the revised Supplementary Information: “During the derivation process, in order to simplify the physical model, we assumed that the photogenerated holes do not need to diffuse a long distance to reach the traps due to the small thickness of NR and the existence of sufficient surface states, thus giving rise to a time-independent constant of B. In this case, the charge accumulation process is an exponential Debye evolution process, which could be regarded as a particular case ($\beta = 1$) of stretched exponential evolution process ($I_{ph} = a \cdot (1 - e^{-(b \cdot P \cdot t)^\beta})$), Phillips, **Rep. Prog. Phys.** 1996, 59, 1133). For the case of the existence of few surface states or a large NR thickness, the photogenerated holes may have to diffuse a long distance to the traps. Therefore, in order to accurately describe the charge accumulation process, an exponent factor β ($0 < \beta < 1$) should be introduced to correct the derived exponential function.”

In addition, to make the formula derivation process more clearly, the detailed derivation process was added in the revised Supplementary Information as Supplementary Scheme 1.

Boundary condition: Irradiation time $t=\infty$
the number of holes captured by the surface states $N_h(\infty)=N$
the number of electrons trapped at the surface state $N_e(\infty)=0$
the number of electrons in the NR $n_e(\infty)=n$
the current of the NR $I(\infty)=D \cdot n_e(\infty)=D \cdot n$, where D is a constant

Boundary condition: Irradiation time $t=0$
the number of holes captured by the surface states $N_h(0)=0$
the number of electrons trapped at the surface state $N_e(0)=N$
the number of electrons in the NR $n_e(0)=n-E \cdot N$, where E is a constant
the current of the NR $I(0)=D \cdot n_e(0)=D \cdot (n-E \cdot N)=I_{dark}$

Irradiation time t
the number of holes captured by the surface states $N_h(t)$
The captured holes and the surface electrons recombine nonradiatively.
the number of electrons trapped at the surface state $N_e(t)=N-N_h(t)$

During the next irradiation time Δt
the number of holes generated by light irradiation $N_{light}=A \cdot P \cdot \Delta t$
where A is a constant and P is the light intensity

During the next irradiation time Δt
the number of holes captured by surface electrons
 $\Delta N_h(t)=B \cdot N_{light} \cdot N_e(t)=B \cdot (A \cdot P \cdot \Delta t) \cdot (N-N_h(t))$ (1)
where B is a constant

Irradiation time $t+\Delta t$
the number of holes captured by the surface states
 $N_h(t+\Delta t)=N_h(t)+\Delta N_h(t)$

$\Delta t \rightarrow 0$,

$$dN_h(t)=B \cdot (A \cdot P \cdot dt) \cdot (N-N_h(t))=A \cdot B \cdot P \cdot (N-N_h(t)) \cdot dt \quad (2)$$

$$\left[\frac{dN_h(t)}{N-N_h(t)} \right] = \left[A \cdot B \cdot P \cdot dt \right]$$

Irradiation time t

$$\ln(N-N_h(t)) = -A \cdot B \cdot P \cdot t + C, \text{ where C is a constant}$$

From the boundary condition,

$$N_h(t) = N \cdot (1 - e^{-A \cdot B \cdot P \cdot t}) \quad (3)$$

$$N_e(t) = N \cdot e^{-A \cdot B \cdot P \cdot t} \quad (4)$$

the number of electrons in the NR

$$n_e(t) = n - E \cdot N_e(t) = n - E \cdot N \cdot e^{-A \cdot B \cdot P \cdot t} \quad (5)$$

the current of the NR

$$I(t) = D \cdot n_e(t) = D \cdot (n - E \cdot N \cdot e^{-A \cdot B \cdot P \cdot t}) \quad (6)$$

the light current of the NR

$$I_{ph} = I(t) - I_{dark} = D \cdot (n - E \cdot N \cdot e^{-A \cdot B \cdot P \cdot t}) - D \cdot (n - E \cdot N) = D \cdot E \cdot N \cdot (1 - e^{-A \cdot B \cdot P \cdot t})$$

Simplifying formula

$$I_{ph} = a \cdot (1 - e^{-b \cdot P \cdot t}), \text{ where a and b are constants} \quad (7)$$

a represents the saturation value of I_{ph}

$b \cdot P$ represents the rate constant of MPD approaching the saturation value

Supplementary Scheme 1. Schematic of the formula derivation process of the exponential-association photoelectric conversion law.

(3) All the explanation is based on the existence of surface trap states. However, the verification and investigation on the surface states are absent either experimental or theoretically. Where the surface states originate from? Is it dangling bond related? Are the states negatively charged or positively charged? What's the density of the states?

Reply: We thank the reviewer for the constructive suggestion. We note that direct characterization of the surface states is difficult through conventional detection techniques. Therefore, as an alternative, we first confirmed the existence of sulfur vacancies in the CdS NRs by electron paramagnetic resonance (EPR) spectroscopy, and then the surface states induced by oxygen adsorption at sulfur vacancies were further investigated by the first-principles calculation. Supplementary Fig. 8 shows the EPR spectrum of the as-prepared CdS NRs. For comparison, the spectra of CdS powder before and after H₂/Ar annealing were also measured. It is noteworthy that, in contrast to the CdS powder before annealing, both the CdS NRs and annealed CdS powder show an obvious signal at g=2.001, revealing the existence of sulfur vacancies due to the high-temperature evaporation process in H₂/Ar. It is noted that similar signals from sulfur vacancies were also observed in the EPR spectra of other metal chalcogen semiconductors (MoS₂, Zhang et al., *Adv. Funct. Mater.* 2018, 28, 1707578; ZnIn₂S₄, Wang et al., *J. Mater. Chem. A* 2017, 5, 8451).

Supplementary Fig. 8 EPR spectrum of CdS NRs. For comparison, the spectra of CdS powder before and after H₂/Ar annealing at 800 °C for 1 h were also measured.

On the other hand, the first-principles calculation was conducted to study the formation of surface energy states within the band gap of CdS NRs after adsorption of oxygen molecules at sulfur vacancies. Supplementary Fig. 9a and 9b show the optimized configurations of one oxygen molecule-adsorbed CdS surface without and with sulfur vacancy, respectively. It is noteworthy that oxygen molecule has lower adsorption energy ($E_{\text{ads}} = -0.87$ eV) on CdS surface with sulfur vacancy, and the

molecule is preferentially adsorbed close to the site of sulfur vacancy. The adsorption of oxygen molecule at sulfur vacancy can significantly change the electrical structure of CdS. As shown in Supplementary Fig. 9c, a new deep energy state is observed above the valence band, which is generated by the adsorbed oxygen molecule according to the projected density of states (PDOS) spectra. In addition, as discussed in Supplementary Fig. 10, the surface states will be negatively charged after electrical erasing, while they will become the unoccupied empty states after the optical programming.

Supplementary Fig. 9 (a,b) Energetically favorable configurations of one oxygen molecule-adsorbed CdS surface without and with sulfur vacancy, respectively. The adsorption energy E_{ads} is defined by the following equation: $E_{ads} = E_{O_2-cds} - E_{cds} - E_{O_2}$, where E_{O_2-cds} , E_{cds} , and E_{O_2} represent the energy of the full adsorption system, the CdS crystal, and the adsorbed O_2 molecule, respectively. (c) Total density of states (DOS, blue lines) spectra of CdS before and after the oxygen adsorption. In the top, the gray and red lines represent the projected density of states (PDOS) of CdS and O_2 in the adsorption system, respectively. The arrow indicates the surface energy state introduced from the adsorbed oxygen molecule.

Accordingly, we added the Supplementary Fig. 8 and Fig. 9 in the Supplementary Information, and the following discussion was changed in Page 7–8: “When oxygen molecules are adsorbed on CdS NR surface, they tend to fill the sulfur vacancies at the surface, leading to the formation of a series of shallow and deep energy states within the band gap of CdS NRs.⁴⁴⁻⁴⁷ Supplementary Fig. 8 shows the electron paramagnetic resonance (EPR) spectrum of CdS NRs. For comparison, the spectra of CdS powder before and after H_2/Ar annealing were also measured. The signal at $g=2.001$ clearly indicates the existence of sulfur vacancies in the CdS NRs due to the high-temperature evaporation process in H_2/Ar environment. We note that similar signals from sulfur vacancies were also observed in the EPR spectra of other metal chalcogen semiconductors.^{48, 49} Supplementary Fig. 9a and 9b show the optimized configurations of one oxygen molecule-adsorbed CdS surface without and with sulfur vacancy, respectively. It is observed that oxygen molecule has lower adsorption

energy ($E_{\text{ads}} = -0.87$ eV) on CdS surface with sulfur vacancy, and the molecule is preferentially adsorbed close to the site of sulfur vacancy. The adsorption of oxygen molecule at sulfur vacancy can significantly change the electrical structure of CdS. As shown in Supplementary Fig. 9c, a new deep energy state is observed above the valence band, which is generated by the adsorbed oxygen molecule according to the projected density of states (PDOS) spectra.”

As for the density of the surface states, we can roughly estimate it from the light density dependent device performance (Fig. 3a–3e in revised manuscript). It is worth noting that the maximum values of responsivity and detectivity appear at a light intensity of 190–380 nW cm⁻² ($\alpha \approx 1$). At this point, almost all the electrons at storage states have been recombined with photogenerated holes, thereby the device performance can reach the highest. If we assume that the number of surface states is approximately equal to the number of photogenerated electrons in the device channel, the density of surface states could be calculated to be 1.9×10^{12} – 3.7×10^{12} cm⁻² from the photocurrent at light intensity of 190–380 nW cm⁻² based on the following formula: $I_{ph} = \mu \cdot q \cdot V_{DS} \cdot \frac{W+2h}{L} \cdot N$, where μ , q , V_{DS} , W , h , L , and N are the mobility of NR, the elementary charge, the source-drain voltage, the width of NR, the thickness of NR, the length of NR, and density of the surface states, respectively. Accordingly, we added the following discussion in Page 11: “If we assume that the number of surface states is approximately equal to the number of photogenerated electrons in the device channel, the density of surface states could be estimated to be 1.9×10^{12} – 3.7×10^{12} cm⁻² from the photocurrent at light intensity of 190–380 nW cm⁻².”

(4) The carrier accumulation mechanism indicates that the response time would be slow. Authors use pulses light to excite the device and get a relatively faster response. What’s the response time? The response speed is important for the photosensors used in the optical communication application. Does this device have this potential?

Reply: We agree with the reviewer that the accumulation mechanism of the MPD devices will limit the response speed. This may hinder their application in optical communication. However, considering the ultra-high responsivity and detectivity of the MPD devices, they are expected to have greater potential for ultraweak light detection in the fields like astronomical observation, remote sensing, laser ranging, and night vision. The requirement for response speed is not so strict in these applications. From the magnified photoresponse curve in Supplementary Fig. 13b, a rise time and a fall time of 0.12 and 0.7 s, respectively, are obtained in the case of without applying an erase pulse (black curve). The response speed is comparable with those of phototransistors (7.0 and 10.7 s, Gong et al., *Adv. Funct. Mater.* 2016, 26, 6084) and photoconductors (0.56 and 0.32 s, Liu et al., *Nat. Commun.* 2014, 5, 4007)

in previous reports. In order to improve the device sensitivity, an erase pulse of +60 V for 0.23 s was also applied on the CdS NR MPD. In this case, the rise and fall time are estimated to 0.5 and 0.5 s (red curve), respectively. After applying the erase pulse, the OFF current of the CdS MPD is suppressed to be $\sim 10^{-14}$ A, thus giving rise to a high sensitivity of 10^8 .

According to reviewer's comment, we added the Supplementary Fig. 13 in the Supplementary Information, and the following discussion was added in Page 12: "From the magnified photoresponse curve in supplementary Fig. 13, the rise/fall time can be estimated to be 0.12/0.7 and 0.5/0.5 s in the cases of with and without an erase pulse, respectively."

Supplementary Fig. 13 (a) Photoresponse curves of the CdS NR phototransistor measured under pulsed light irradiation (2000 nW cm^{-2}). A fixed gate voltage (corresponding to black curve) or a periodically pulsed gate voltage (corresponding to red curve) was applied during measurements. (b) Enlarged rise and fall edges of the temporal response.

(5) Do any other trap states contribute to the performance? Is surface state the only contributor? If the surface state is the dominant factor, does this mean the thinner or narrower nanoribbons would have better performance?

Reply: It is believed that the surface states are the dominant factor that contributes to the performance. In previous reports, the defect states in the dielectric layer of substrate (Radosavljević et al., *Nano Lett.* 2002, 2, 761) or the water molecules adsorbed on the substrate (Kim et al., *Nano Lett.* 2003, 3, 193) can also act as the trap states. To exclude these possibilities, control experiments were performed by pre-annealing the substrate in H_2/Ar at $1100 \text{ }^\circ\text{C}$ for 2h or passivating the substrate with 10 nm thick Al_2O_3 . The Supplementary Fig. 7a and 7b show the electrical transfer characteristics of the CdS NR phototransistor fabricated on the pre-annealed and passivated substrates, respectively. Note that the memory characteristics of the devices do not show obvious deterioration. In contrast, the storage window of the

device is significantly reduced after Al₂O₃ passivation on the surface of CdS NR (Fig. 2b in manuscript). This result clearly demonstrates that the charge storage sites mainly come from the surface states on the CdS NR. This assumption is further supported by fact that the suspended CdS NR-based MPD shows larger optoelectronic memory window (Fig. 4 in manuscript). In this case, the number of surface storage states could be effectively increased by exposing the bottom surface of the NR.

The NR dimension, especially the thickness of the NR, should be an important factor that determines the device performance. With decreasing thickness, the surface-to-volume ratio will be increased, leading to enhanced device performance. However, considering the fact that the light absorption coefficient of CdS at 500 nm wavelength is around $5 \times 10^6 \text{ m}^{-1}$ (Senthil et al., **Semicond. Sci. Technol.** 2002, 17, 97), this means that the light penetration depth in CdS NR is around 200 nm. Further decrease of the NR thickness below 200 nm will cause the decrease of light absorption. Therefore, the optimum NR thickness should be 100–200 nm. In our work, the thickness of CdS NRs is about 100–150 nm, so the devices show relatively consistent device performance (Supplementary Fig. 12). In contrast to the thickness, the influence of NR width could be neglected due to the small change of specific surface area by NR narrowing. For example, for the CdS NR in our device, if the width of CdS NR decreases from 2 to 1 μm , the specific surface area will only increase slightly from 0.021 to 0.022 nm^{-1} . In addition to the NR dimension, we have demonstrated that the photodetection performance of the MPD device can be effectively improved by controlling the exposed surface area of CdS NR (Fig. 4 in manuscript). The optoelectronic memory window, the voltage responsivity, and the subthreshold swing of CdS NR can be improved by fabricating MPD device based on the suspended CdS NR.

Accordingly, we added the Supplementary Fig. 7 in the Supplementary Information, and the following discussion was added in Page 6–7: “The Supplementary Fig. 7a and 7b show the electrical transfer characteristics of the CdS NR phototransistor fabricated on the pre-annealed and surface passivated substrates, respectively. Note that the memory characteristics of the devices do not show obvious deterioration. This result excludes the possibilities of defect states on the dielectric layer or adsorbed water molecules on the substrate as storage mediums.”

Supplementary Fig. 7 Electrical transfer characteristics of the CdS NR phototransistor fabricated on the pre-treated substrates. The substrate was (a) pre-annealed in H_2/Ar at 1100 °C for 2h or (b) passivated with 10 nm thick Al_2O_3 .

Response to reviewer #3:

Reviewer's comments:

(1) The work presented by the Author has a wider applicability because of its hybrid nature and have a two fold approach and is therefore quite interesting for the researcher.

Reply: We would like to thank reviewer for the positive comment on our manuscript.

(2) The work seems to be original, however, characterization of CdS nano ribbons has not been given in the submitted manuscript.

Reply: We thank the reviewer for the valuable comment. Following the reviewer's suggestion, we provided the detailed characterizations on CdS NRs in the Supplementary Information (Supplementary Fig. 1). Accordingly, we added the following discussion in Page 4 in revised manuscript: "The CdS NRs exhibit a single-crystalline wurtzite structure with growth orientation of [001]. The width and thickness of the NRs are 0.4–5 μm and 100–150 nm, respectively, while the length of the NRs is up to several hundreds of micrometers (Supplementary Fig. 1)." The experimental details were added in Page 16 in revised manuscript: "Scanning electron microscope (SEM, Hitachi SU5000), X-ray diffraction (XRD, PANalytical Empyrean), atomic force microscope (AFM, Cypher S) and transmission electron microscope (TEM, FEI Tecnai G2 F20 S-TWIN) were used to characterize the morphologies and structures of CdS NRs."

Supplementary Fig. 1 Characterizations of CdS NRs synthesized by physical vapor deposition. (a) SEM image, (b) EDS spectrum, (c) XRD pattern of the CdS NRs. (d) AFM three-dimensional (3D) morphology image, (e) TEM image, and (f) SAED

pattern of an individual CdS NR.

Supplementary Fig. 1a shows the SEM image of the as-prepared CdS NRs, revealing that a host of CdS NRs with a width of 0.4–5 μm and a length of up to hundreds of micrometers were synthesized. From the EDS spectrum and XRD pattern in Supplementary Fig. 1b and 1c, respectively, it can be deduced that the NRs are hexagonal CdS. AFM image measurement indicates a thickness in the range of 100–150 nm for the CdS NRs (Supplementary Fig. 1d). Supplementary Fig. 1e and 1f show the TEM image and corresponding SAED pattern, respectively, of a single CdS NR. The CdS NR possesses single-crystalline wurtzite structure with a growth orientation of [001]. The top and bottom surfaces of the CdS NR are deduced to be (100) facets.

(3) Photoelectric storage Mechanism has been explained, however, a control experiment may be carried out as a function of sulfur vacancies during the synthesis.

Reply: We thank the reviewer for the constructive suggestion. To investigate the influence of sulfur vacancies, we measured the device performance of an individual CdS NR by tuning the amount of sulfur vacancies with the electron-beam irradiation. This approach can effectively avoid the error induced by performance fluctuation between different NRs. In previous works, electron-beam irradiation has been demonstrated to be an effective approach to introduce chalcogen vacancies on the surface of metal chalcogen semiconductor (Lin et al., **Nat. Commun.** 2015, 6, 6736; Sutter et al., **Nano Lett.** 2016, 16, 4410; Moody et al., **Phys. Rev. Lett.** 2018, 121, 057403).

To increase the amount of sulfur vacancies, the CdS NR MPD was uniformly irradiated with an electron beam in a SEM with 4×10^6 electrons/ μm^2 (30 kV accelerating voltage, 223 pA current, 9.9 nm spot size for scanning). Supplementary Fig. 18a shows the electrical transfer characteristics of the CdS NR MPD after 0, 20, and 40 s of electron-beam irradiation. In addition, electrical-optical hybrid storage windows and pure electrical storage windows of the MPD are further extracted from the transfer characteristics (Supplementary Fig. 18b). We note that, after the electron-beam irradiation, the proportion of the electrical storage window in the total storage window has increased significantly, implying an enhanced electrical storage capability of the MPD. This result could be attributed to the excess shallow energy states generated after electron-beam irradiation. As discussed in Supplementary Fig. 10, the electrical and optical storage capabilities are mainly associated with the shallow and deep energy states, respectively. Due to the small energy barrier between the shallow energy states and conduction band of CdS NR, the electrical storage capability is preferentially improved when the densities of shallow and deep energy

states are simultaneously increased. The strong electrical storage will interfere with the optical storage process, thus limiting the performance of MPD for ultraweak light detection. Therefore, in the MPD, rationally controlling the surface state density of the semiconductor nanostructures is crucial for achieving a high performance for weak light detection.

Accordingly, we added the Supplementary Fig. 18 in the Supplementary Information, and the following discussion was added in Page 13–14: “However, further increasing the density of surface states would preferentially increase the electrical memory window, leading to degeneration of the device performance. Electron-beam irradiation is an effectively approach to introduce the chalcogen vacancies on the surface of metal chalcogen semiconductor.⁵⁷⁻⁵⁹ From Supplementary Fig. 18, it is observed that the proportion of the electrical storage window in the total storage window increases significantly upon the electron-beam irradiation, manifesting an enhancement of electrical storage capability for the MPD. This result could be attributed to the excess shallow energy states generated after electron-beam irradiation. As discussed in Supplementary Fig. 10, the electrical and optical storage capabilities are mainly associated with the shallow and deep energy states, respectively. Due to the small energy barrier between the shallow energy states and conduction band of CdS NR, the electrical storage capability is preferentially improved when the densities of shallow and deep energy states are simultaneously increased. The strong electrical storage will interfere with the optical storage process, thus limiting the performance of MPD for ultraweak light detection. Therefore, in a MPD, rationally controlling surface state density of the semiconductor nanostructures is crucial for achieving a high performance for weak light detection.”

Supplementary Fig. 18 (a) Electrical transfer characteristics of CdS MPD after 0, 20, and 40 s of electron-beam irradiation. The CdS MPD was uniformly irradiated with an electron beam in a SEM with 4×10^6 electrons μm^{-2} (30 kV accelerating voltage, 223 pA current, 9.9 nm spot size for scanning). The device was measured in the dark with

(solid line) and without (dash line) the light pre-irradiation (100 nW cm^{-2}) for 10 s. (b) Statistical histograms of memory windows extracted from (a).

(4) It would be appreciable and helpful for the author to carry out the conductive AFM studies in the Low and high resistance state of the CdS NRs after and before the light illumination for further understanding the mechanism.

Reply: We thank the reviewer for the valuable suggestion. Following reviewer's suggestion, we have measured the surface potentials of CdS NR before and after turning off the light by using Kelvin probe force microscope (KPFM) (Supplementary Fig. 6). We note that the surface potential of CdS NR only decreases slightly after turning off the light, indicating that the NR can maintain at low resistance state (ON state) even without light irradiation. This result is consistent with the persistent photoconductivity behavior observed for the MPD device (Fig. 1d and Supplementary Fig. 5). Accordingly, we added the Supplementary Fig. 6 in the Supplementary Information, and the following discussion was added in Page 6: "Furthermore, Kelvin probe force microscope (KPFM) was utilized to measure the surface potential of CdS NR (Supplementary Fig. 6). We note that the surface potential of the NR only decreases slightly after turning off the light, implying that the carrier concentration changes little after light irradiation. This result is consistent with the electrical measurements on the phototransistor."

Supplementary Fig. 6 (a) Top: 2D topography image of the CdS NR. Middle: Kelvin potential image of CdS NR measured before turning off the light. Bottom: Kelvin potential image of the CdS NR measured after turning off the light. (b) Line profiles of Kelvin voltage extracted from the potential images measured before (red) and after turning off the light (blue), respectively.

(5) Abstract is clear and understandable.

Reply: We thank reviewer for the positive comment.

(6) May be accepted after minor suggested revision.

Reply: Following reviewer's suggestions, we have revised the manuscript.

REVIEWERS' COMMENTS:

Reviewer #1 (Remarks to the Author):

Authors revised the manuscript well. It can be published in Nature Communications.

Reviewer #2 (Remarks to the Author):

I have no more questions on the revised manuscript and recommend for publication.

Reviewer #3 (Remarks to the Author):

Comment 1: The author has tried to answer the queries in the revised manuscript and is found to be successful in addressing all of them. Specially the experiment to understand the effect of sulfur vacancy have turned out to be signature for increasing the effect of MPD for ultra weak light detection.

Comment 2: The revised manuscript may be accepted as it is.

Reviewers' comments

Reviewer #1:

Authors revised the manuscript well. It can be published in Nature Communications.

Reply: We thank reviewer for the very positive comment on our revised manuscript.

Reviewer #2:

I have no more questions on the revised manuscript and recommend for publication.

Reply: We would like to thank reviewer for the recommendation on our revised manuscript.

Reviewer #3 (Remarks to the Author):

Comment 1: The author has tried to answer the queries in the revised manuscript and is found to be successful in addressing all of them. Specially the experiment to understand the effect of sulfur vacancy have turned out to be signature for increasing the effect of MPD for ultra weak light detection.

Comment 2: The revised manuscript may be accepted as it is.

Reply: We would like to thank reviewer for the positive comments on our revised manuscript.